# The AML cellular state space unveils *NPM1* immune evasion subtypes with distinct clinical outcomes

Henrik Lilljebjörn [1] ✉, Pablo Peña-Martínez [1], Hanna Thorsson [1], Rasmus Henningsson[1], Marianne Rissler[1], Niklas Landberg [1], Noelia Puente-Moncada[1], Sofia von Palffy [1], Vendela Rissler[1], Petr Stanek [1], Jonathan Desponds [2,3], Xiangfu Zhong [4], Gunnar Juliusson[5], Vladimir Lazarevic [5], Sören Lehmann[4], Magnus Fontes[2], Helena Ågerstam[1,6], Carl Sandén [1], Christina Orsmark-Pietras [1,6] & Thoas Fioretos [1,6,7] ✉

Acute myeloid leukemia is a genetically and cellularly heterogeneous disease. We characterize 120 AMLs using genomic and transcriptomic analyses, including single-cell RNA sequencing. Our results reveal an extensive cellular heterogeneity that distorts the bulk transcriptomic profiles. Selective examination of the transcriptional signatures of >90,000 immature AML cells identifies four main clusters, thereby extending current genomic classification of AML. Notably, *NPM1*-mutated AML can be stratified into two clinically relevant classes, with *NPM1*^class1 associated with downregulation of MHC class II and excellent survival following hematopoietic stem cell transplantation. *NPM1*^classII is instead associated with resistance to allogeneic T cells in an ex vivo co-culture assay, and importantly, dismal survival following hematopoietic stem cell transplantation. These findings provide insights into the cellular state space of AML, define diagnostic entities, and highlight potential therapeutic intervention points.

Acute myeloid leukemia (AML) is a highly heterogeneous malignancy characterized by multiple disease-driving molecular alterations of diagnostic, prognostic, and therapeutic importance[1]. The disease is driven by rare populations of leukemia stem cells (LSCs), typically contained within the most immature cell population, which generate the more differentiated and dysfunctional bulk of blast cells observed at diagnosis. The AML stem cells are, however, insensitive to most therapeutic strategies and frequently escape the treatment and cause relapse[2,3]. Despite the fact that an increasing number of targeted therapies have become available during recent years, intensive chemotherapy still provides the backbone treatment strategy, which is

associated with significant toxicities and a high risk of relapse[1,4,5]. The 5-year overall survival (OS) of AML is 24%. However, the age at diagnosis is an important factor; younger patients (<65 years) have a 5-year OS of 50% while the corresponding survival for elderly patients (>65 years) is less than 10%[4,5].

Adjusting the treatment based on the presence of specific molecular alterations has significantly improved the outcome for specific subtypes of AML, such as acute promyelocytic leukemia (APL) and core-binding factor AML (CBF-AML), where the 5-year OS is over 70% with modern protocols[6,7]. Thus, a precise knowledge of the underlying subtypes in AML and how molecular vulnerabilities are distributed

[1]Division of Clinical Genetics, Department of Laboratory Medicine, Lund University, Lund, Sweden. [2]Institut Roche, Boulogne-Billancourt, France. [3]Symphogen, Ballerup, Denmark. [4]Department of Medicine Huddinge, Center for Hematology and Regenerative Medicine, Karolinska Institute, Huddinge, Sweden. [5]Department of Hematology, Oncology and Radiation Physics, Skåne University Hospital, Lund, Sweden. [6]Department of Clinical Genetics, Pathology, and Molecular Diagnostics, Skåne University Hospital, Region Skåne, Lund, Sweden. [7]Clinical Genomics Lund, Science for Life Laboratory, Lund University, Lund, Sweden. ✉e-mail: henrik.lilljebjorn@med.lu.se; thoas.fioretos@med.lu.se

across the subtypes and their stem cell populations can provide a guide for optimal implementation of new and coming AML treatment regimens.

AML was the first cancer type to become fully characterized by whole-genome sequencing (WGS), and pioneering studies by the Cancer Genome Atlas Project (TCGA) enabled the detection of multiple new disease-driving alterations[8,9]. Subsequent systematic analyses of co-occurring and mutually exclusive alterations in large AML cohorts have been used to define distinct genomic subtypes[10,11]. Several, but not all, of these molecular subtypes have been included in current diagnostic guidelines, available from the World Health Organization (WHO) and the International Consensus Classification (ICC)[12,13]. The most common subtypes of AML are characterized by *NPM1*-mutations (30%), myelodysplasia-related mutations (AML-MR; defined by the presence of mutation in one of 8 or 9 genes; 20%), and *TP53*-mutation or complex karyotype (15%) [10].

Using bulk RNA-sequencing (RNA-seq), we and others have shown that the genetic subtypes of acute lymphoblastic leukemia (ALL) display characteristic transcriptional programs, driven by the underlying disease-causing molecular alterations, which result in distinct gene expression profiles of clinical and prognostic importance[14–16]. Intriguingly, only a subset of the suggested molecular subtypes of AML have been described to be associated with distinct gene expression profiles; this includes several gene fusions, *NPM1* mutations, and biallelic *CEBPA* mutations[17–24]. A stronger correlation between molecular subtype and gene expression-based clusters was observed when more advanced clustering techniques were used[24]. However, the feature with the strongest correlation to unsupervised gene expression-based clusters has been the apparent maturation stage of the leukemic cells[9,19,21]. Thus, the underlying cause for the global gene expression differences between the majority of AML samples has yet to be elucidated.

Recent technological advancements have permitted studying AML using single-cell RNA-sequencing (scRNA-seq). This has revealed that AML samples have a variable cell composition, with differentiated AML cells expressing immunomodulatory factors that suppress T cells, and that cell fractions with distinct subclonal mutations have distinct gene expression profiles[25–27]. In addition, in-depth analyses of small cohorts using low-throughput methods with single-cell mutation calling have revealed the molecular consequences of specific mutations in different cell types, as well as the gene expression differences in immature AML cells between diagnosis and relapse [28–30].

In this work, we show that unbiased clustering of the bulk gene expression data is mainly driven by the presence of expression profiles matching normal cells. However, further delineation of the cellular composition with single-cell RNA sequencing unmasks this diversity within the AML cellular state space. Importantly, we show that *NPM1*-mutated AML can be divided into two separate classes based on the distinct expression profiles of the most immature AML cells: *NPM1*[class I] and *NPM1*[class II], with significantly different responses to hematopoietic stem cell transplantation (HSCT). We propose that a difference in T cell sensitivity underlies the survival difference following HSCT, and that this is linked with different immune evasion strategies by the two subtypes, where *NPM1*[class I] exhibits a downregulation of MHC class II components, and *NPM1*[class II] is associated with increased resistance to allogeneic T cells.

## Results
### Integrative sequencing reveals that AML bulk gene expression profiles are highly affected by cell heterogeneity
To identify subtypes of AML, based both on mutational patterns and gene expression profiles, we performed integrative sequencing analysis on 120 matched tumor-normal AML cases, representing 112 unique patients. This analysis consisted of whole exome sequencing (WES, 120 cases), mate-pair whole genome sequencing (MP-WGS, 98 cases), RNA-seq (120 cases), and single cell multimodal sequencing

(including scRNA-seq, single cell antibody-derived tag-sequencing – scADT-seq – and single cell mutation calling on scRNA-seq – scRNA-mut-seq; 38 cases; Supplementary Data 1). Bulk sequencing (WES, MP-WGS, and RNA-seq) was performed on material extracted from unsorted white blood cells, and single-cell multimodal sequencing was performed on live-sorted mononuclear cells (MNCs), all from either peripheral blood (PB) or bone marrow (BM). The bulk data on somatic variants (SNVs, indels, copy number changes, and fusion genes) was used to categorize the cases into thirteen molecular subgroups[10] (Fig. 1a). The mutational landscape was similar to what has previously been described in large scale studies of AML[9,10,31] (Supplementary Fig. 1), although our consecutive series of AML contained a relatively high proportion of AML with myelodysplasia-related gene mutations (AML-MR, 27%, compared to 13% in TCGA, 18% in the Papaemmanuil cohort, and 27% in Beat-AML).

Unsupervised hierarchical clustering of gene expression data from 120 AMLs revealed seven distinct clusters (clusters 1–7; Fig. 1b). Notably, four AMLs with gene fusions affecting *KAT6A* (*KAT6A::EP300* and *KAT6A::NCOA2*) or *KMT2A* (*KMT2A::MLLT3*) in combination with *TP53*-mutations formed a small subcluster (cluster 4a) with a uniform gene expression pattern, potentially identifying a subtype. AMLs that have previously been described to display uniform bulk gene expression patterns, such as those with *PML::RARA*, *RUNX1::RUNX1T1*, and *CBFB::MYH11* fusion genes, as well as AML with *NPM1* mutations[9,17–21], mostly clustered together. However, neither cases with *CBFB::MYH11* nor cases with *NPM1* mutations formed single clusters encompassing all cases with the respective aberration. AMLs with *CBFB::MYH11* were divided into two distinct expression clusters. AMLs with *NPM1* mutations were also divided into two expression clusters, but with additional cases present outside the two main clusters. These divisions were the result of a difference in expression of four prominent gene clusters (clusters A-D; Fig. 1b). In fact, variable expression of gene clusters A-D was the major factor driving the unsupervised hierarchical clustering for all samples, since the expression level of these gene clusters were markedly different between the seven sample clusters (Fig. 1b). Interestingly, these four gene clusters were highly enriched for cell type markers from platelets, fibroblasts, erythroid precursor cells, and neutrophils, thus possibly representing specific cell types in the sequenced bulk samples, rather than intrinsic features of the AML blast populations (Supplementary Fig. 2). In support of this notion, the expression of genes from clusters A-D was markedly reduced in samples that were sorted to contain only myeloid (CD33+/CD19-/CD3-) mononuclear cells (Supplementary Fig. 3a). Notably, AML gene expression datasets from TCGA[9] and Beat-AML[31] exhibited the same type of co-regulation for genes in clusters A-D, with high variability between samples, as in our dataset (Supplementary Fig. 3b-c). Collectively, we conclude that the bulk gene expression profiles of AML samples are highly affected by their cellular contents, and that such strong deviations have the potential to conceal the presence of any subtype-associated expression profiles within the AML blast cells.

Given the limitation of bulk RNA gene expression analysis to classify and define the cellular and biological characteristics of AML, we next performed single-cell multimodal sequencing of a subset of 38 AML samples together with eight normal bone marrow (NBM) samples. The AML samples represented the subgroups *NPM1*-mutated, AML-MR, *TP53*-mutated, *CBFB::MYH11*, *RUNX1::RUNX1T1*, AML without class-defining mutations, and AML meeting criteria for two subgroups. To determine which cells constituted the leukemic clone in each sample, we developed a mutation calling assay (scRNAmut-seq), allowing us to define the status of 1-4 selected AML mutations per sample (Supplementary Fig. 4; Supplementary Data 2). In total, this approach successfully provided genotyping information for 91,727 of the 196,417 cells from AML samples (47%; Supplementary Fig. 5).

To decipher the cellular contents of the AML samples, all cells were projected onto a reference k nearest neighbor force-layout graph

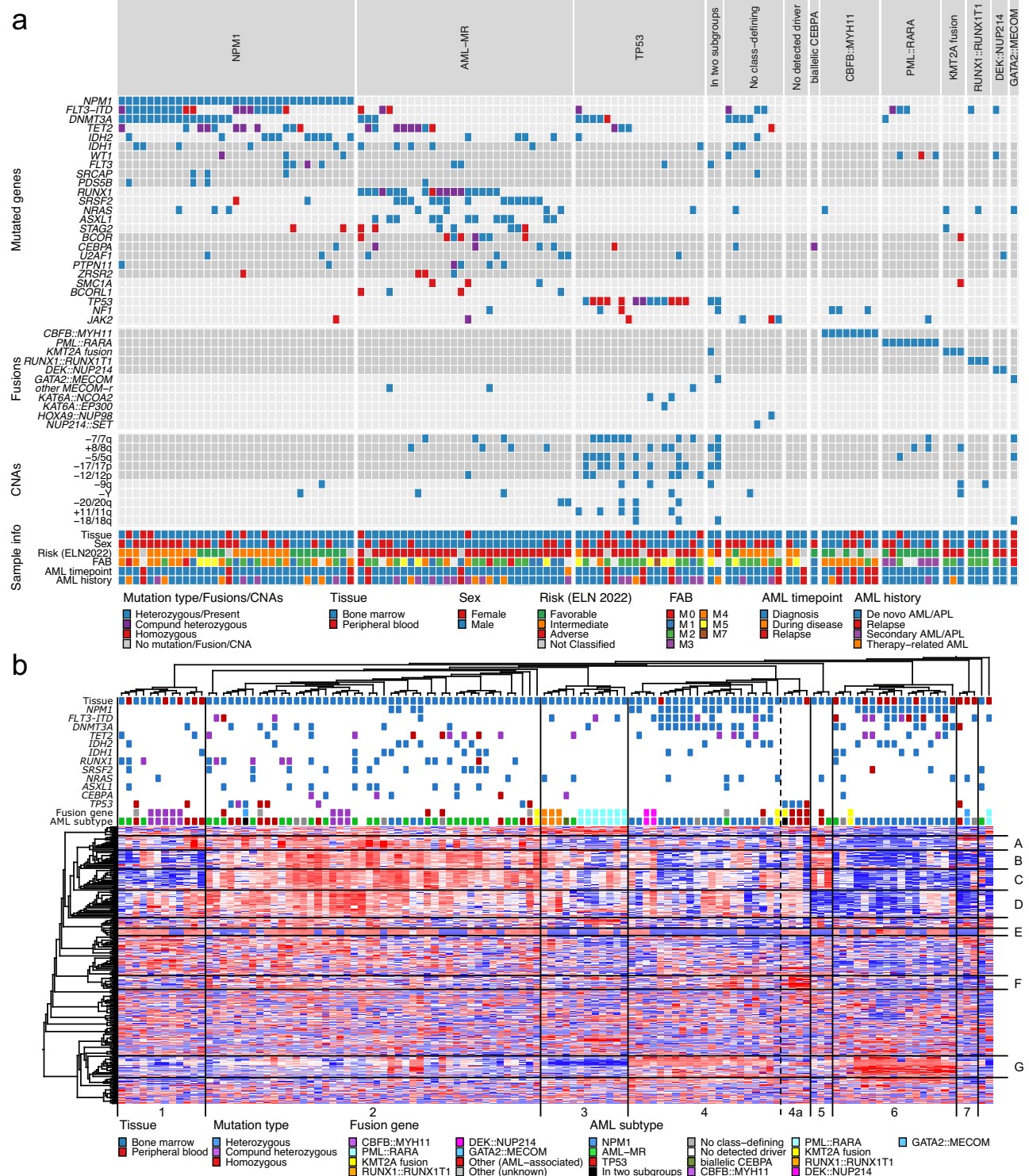

**Fig. 1 | Integrative bulk sequencing and multimodal single cell sequencing of AML. a** Genetic alterations and sample information for the 120 AML samples included in the analyzed cohort. Each column represents a single sample. Samples are classified and arranged according to recently proposed subtype definitions[10]. Mutational status for the most common recurrently mutated genes (mutated in three or more patients) is visualized together with the presence of clinically relevant or recurrent gene fusions, the most common copy number aberrations (CNAs) detected, and selected clinical information. **b** Hierarchical clustering of the 120 AML samples based on gene expression of the 371 most variable genes (threshold set at 0.45 of the highest standard deviation). Columns represent samples and rows represent genes. The presence of the most common mutations and fusion genes, as well as selected clinical information, is marked in colors above the heatmap. Based on the dendrogram above the heatmap, lines denoting separated clusters and dashed lines indicating subclusters have been added to the plot. Sample clusters are marked with numbers (1–7) below the heatmap. Based on the left-side dendrogram, distinct gene clusters are marked with letters to the right of the heatmap. ELN2022, risk group according to the 2022 European LeukemiaNet recommendations; FAB classification according to the French-American-British system, APL acute promyelocytic leukemia. Source data are provided as a Source Data file.

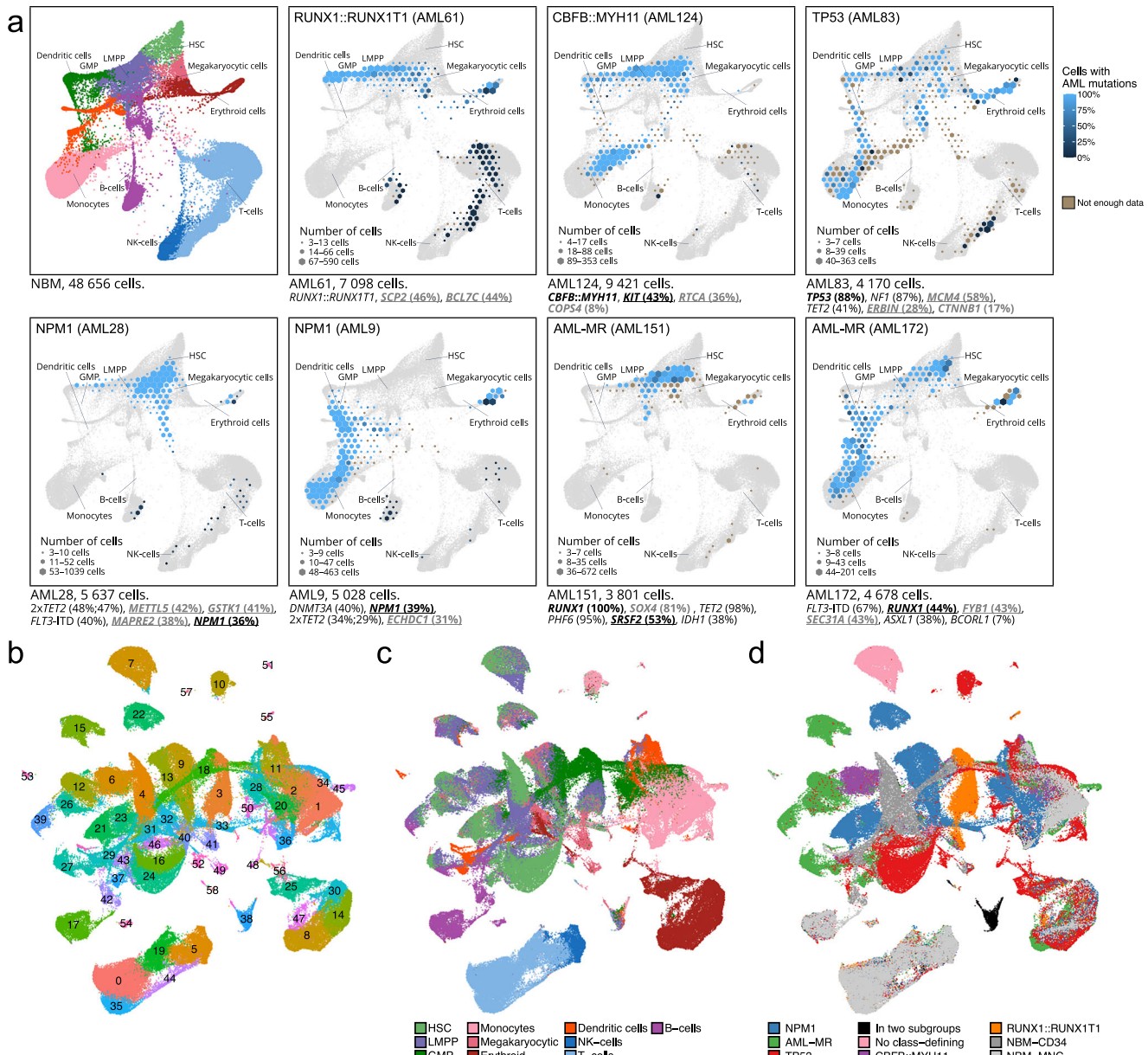

**Fig. 2 | Visualization of AML single cells by UMAP and projection onto a reference NBM knn force graph. a** A knn force graph was constructed from 48,656 normal bone marrow (NBM) cells (top left) representing eight samples. Cells from individual AML samples (indicated in blue) were projected onto the reference NBM knn force graph (indicated in gray). The number of cells projected onto a region is indicated by the size of each hexagon and the proportion of mutated cells in that hexagon is indicated by shade of blue. Hexagons with too few genotype reads are marked in brown (three or fewer reads). The genomic subtype of each sample is indicated in the top left corner. Sample identifier and number of cells are indicated below each plot. A selection of genes mutated in each sample is presented below each plot, with the variant allele frequency indicated by whole exome sequencing denoted in parentheses. Genes targeted by single cell mutation detection using an in-house PCR are indicated in bold. Genes determined to have heterozygous mutations are underlined; these were used for inferring the proportion of mutated cells. Genes with presumed passenger mutations are denoted in gray. **b–d** UMAP representation of 245,073 cells from 38 AML samples and 8 NBM samples. **b** Cells are colored based on graph-based clustering, which identified 58 distinct clusters of cells within the data. **c** Cells are colored by cell type based on classification using a reference dataset of NBM cells. **d** Cells are colored by AML subtype or NBM sample type. Notably, all AML samples were associated with a distinct cluster of cells outside the regions occupied by cells from NBM samples. The majority of cells in these AML-specific clusters were classified as hematopoietic stem cells (HSC) or lymphoid-primed multipotent progenitors (LMPP) based on the reference dataset. The cells in the AML-specific clusters were therefore reclassified as AML immature. GMP Granulocyte-monocyte progenitors. Source data are provided as a Source Data file.

(knn force graph) constructed from NBM samples (Fig. 2a, Supplementary Figs. 6–8). This revealed that the two subtypes defined by gene fusions, *RUNX1::RUNX1T1* and *CBFB::MYH11*, were associated with relatively uniform cellular patterns compared to the other genetic subtypes (Supplementary Figs. 6–8). Both these subtypes contained immature cells resembling LMPP that appeared to mature towards Granulocyte-monocyte progenitors (GMP). However, the three cases

with *CBFB::MYH11* also contained a fraction of monocytic cells whereas the three cases with *RUNX1::RUNX1T1* instead contained a fraction of erythroid cells (Fig. 2a, Supplementary Figs. 6, 8). Interestingly, the remaining subtypes showed a substantial heterogeneity with regards to the maturation stages of the mutated cells, even within the same genomic subtype (Fig. 2a, Supplementary Figs. 6–8). For example, several AMLs from the *TP53*, *NPM1*, and AML-MR subtypes contained

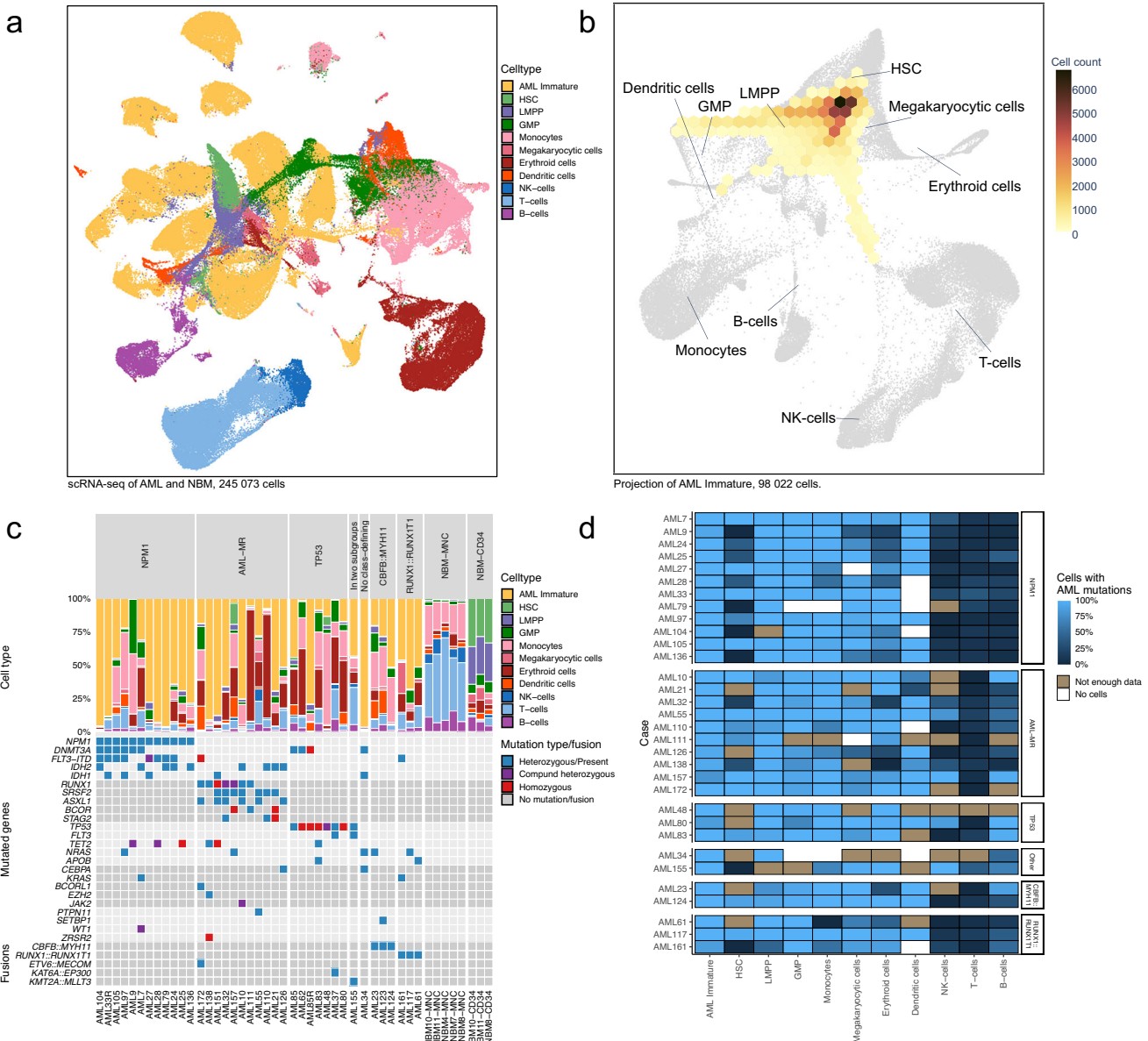

**Fig. 3 | Single cell RNA-seq analyses of 38 AML and 8 NBM samples. a** UMAP representation of 245,073 cells from 38 AML samples and 8 normal bone marrow (NBM) samples. Cells are colored by cell type based on classification using a reference dataset of NBM cells. Clusters of cells from AML samples where a large proportion of cells was classified as hematopoietic stem cells (HSC) or lymphoid-primed multipotent progenitors (LMPP), but did not fully match the gene expression of the corresponding NBM cells, were classified as AML immature. **b** Knn-force plot projection of 98,022 cells classified as AML immature from 38 AML samples, confirming that these cells are most similar to immature NBM cells. The number of cells projected over each hexagon is indicated by the color scale. Only hexagons covered by more than 100 cells (0.1% of the grand total) are displayed in the plot.

**c** Barplot depicting the proportion of cell types in each sample for 245,073 cells from 38 AML samples and 8 NBM samples. Mutated genes and fusion genes present in the samples are indicated below each bar. **d** Heatmap describing the inferred frequency of cells carrying confirmed somatic mutations for each cell type as determined by single cell RNA mutation calling (scRNAmut-seq) on 163,789 cells from 32 AML samples. Brown regions indicate that there were too few reads to infer a mutation frequency (i.e., three or fewer genotype reads). White indicates cell types not present in the current sample. Six samples with too few informative reads were excluded from this analysis. GMP Granulocyte-monocyte progenitors. Source data are provided as a Source Data file.

mainly cells resembling mature monocytes, whereas other cases of the same subtype contained mainly cells resembling immature or a mixture of immature and mature cells (Fig. 2a, Supplementary Figs. 6–8), suggesting the presence of significant biological differences beyond current genomic classification schemes.

To study the cellular contents of the AML samples further, the cells were clustered using a graph-based algorithm (Fig. 2b), a technique that has previously been employed to identify distinct cell types, including the most primitive cells, in an AML context[32]. In addition, the cell type for each cell was classified using a NBM reference (Fig. 2c).

Notably, all AML samples were found to contain a distinct cluster of cells substantially different from any cell type present in normal bone marrow, as indicated by spatial separation in the UMAP (Fig. 2d). Cell type classification indicated that the cells in these clusters were most similar to immature cells such as hematopoietic stem cells (HSC) and lymphoid-primed multipotent progenitors (LMPP). These cells were assumed to constitute immature AML blasts and were designated AML immature (Fig. 3a, Supplementary Fig. 9). The AML immature cells were projected onto the NBM reference knn force graph, which confirmed that their closest counterparts in normal BM were the most

immature cells (Fig. 3b). In addition, immunophenotyping by scADT-seq showed marker expression compatible with LSCs, i.e. CD34+CD38-/low for CD34-positive AMLs and distinct expression profiles of CD33, CD45RA, CD117, CD123, and IL1RAP, suggesting that this cell population harbors the LSCs (Supplementary Fig. 10).

The cell classifications indicated that the cellular contents of the AML samples were highly heterogeneous (Fig. 3c), in concordance with the previous bulk gene expression analysis and single-cell projections. The proportion of AML immature cells varied between 1-97% of each sample (Fig. 3c). This population was largest in *NPM1*-positive AMLs (median size 74%) and lowest in the *TP53*-mutated AMLs (median 16%).

By combining the AML mutation genotyping information and cell classifications, we could infer the proportion of cells with AML mutations within each cell type for 32 of the samples (Fig. 3d). This analysis highlighted that practically 100% of the cells classified as AML immature in each sample harbored AML mutations (Fig. 3d), as did most other cell types from the myeloid lineage. Thus, while the AML immature cells showed the most distinct change in gene expression, practically all cells in the myeloid lineage were part of the leukemic clone. AML mutations were also prevalent in B cells and NK cells (but not T cells) in a small subset of AMLs from the AML-MR or *TP53* subtypes, indicating a less restrictive lineage block or the presence of mutated pre-leukemic cells with intact lymphoid potential, at least in individual cases (Fig. 3d).

In summary, we show that the bulk AML gene expression profiles are driven by clusters of specific mature cell type markers that are likely to represent the variable cell content between samples. This was confirmed by scRNA-seq, which also unveiled an unanticipated level of cellular heterogeneity even within distinct subtypes. Importantly, however, all AML samples were found to contain a subset of immature cells, harboring AML mutations, with an expression profile distinct from any cell type present in healthy BM. Thus, the gene expression information provided by single-cell gene expression analysis allows precise characterization of the molecular consequences of AML within the detected cell types.

## Distinct gene expression profiles in immature AML cells identify subtypes and extend current genomic classifications

The cells classified as AML immature represented distinct cell clusters within the AML samples with no counterpart in normal mononuclear or CD34-enriched bone marrow samples. Notably, they exhibited the closest gene expression similarity to HSC and LMPP cells, and displayed immunophenotypic markers compatible with LSC, indicating their immatureness (Fig. 3b, Supplementary Fig. 10). Each sample was associated with a distinct cluster of such AML immature cells. We therefore hypothesized that subtype-specific gene expression profiles could be distinguished better within isolated AML immature cell populations, without interfering expression profiles from more mature cells. Thus, the average gene expression levels were calculated for the AML immature cell cluster from each sample. Hierarchical clustering analysis of the 361 most variable genes between samples from the averaged gene expression revealed four distinct sample clusters with several subclusters (Fig. 4a). The first cluster (C1) contained only samples from the AML-MR and *TP53* subtypes. This cluster was further subdivided into three subclusters; one cluster (C1a) contained six samples from the AML-MR subtype, which were all samples in this analysis having mutations in *IDH1* or *IDH2* in combination with MDS-related mutations. Notably, clusters C1b and C1c contained samples from the AML-MR and *TP53*-mutated subtypes in equal proportions, indicating molecular similarities between these two subtypes.

Cluster C2 encompassed all cases with fusions affecting the core-binding factor (CBF) complex, i.e. three cases with *CBFB::MYH11* and three cases with *RUNX1::RUNX1T1*. This cluster was further divided into subclusters in accordance with the fusion present (C2a and C2b). Cluster C3 contained all twelve *NPM1*-mutated samples and the single

sample without class-defining lesions. Eleven of the *NPM1*-positive samples were subdivided into two distinct clusters of five and six samples each (C3a and C3b). Lastly, cluster C4 contained two samples with *KAT6A::EP300* and *KMT2A::MLLT3* fusion genes together with *TP53* mutations, which also clustered together based on bulk gene expression data. Thus, compared to hierarchical clustering based on bulk RNA gene expression data, this analysis highlighted gene expression profiles that generally align with expected class-defining genetic features. For example, all samples harboring an aberration known to be associated with a distinct gene expression profile (i.e., *NPM1*, *CBFB::MYH11*, and *RUNX1::RUNX1T1*) clustered according to their genotype. In addition, samples harboring *CBFB::MYH11* and *RUNX1::RUNX1T1* clustered closely together. This aligns well with the known biology since both these fusions affect components of the hetero-dimeric CBF protein-complex and function as dominant repressors of the native CBF transcription factor [33].

Interestingly, the *NPM1*-positive samples that formed the separate subclusters C3a and C3b were also associated with two different sub-clusters of AML immature cells in the UMAP visualization of single cells (Fig. 4b, Supplementary Fig. 11a). Thus, in addition to the distinct *NPM1*-associated expression profile, this sensitive analysis highlighted a subtle gene expression difference among the most immature AML cells in *NPM1* positive AML. The two subclusters were also associated with different cell distributions. Cluster C3a contained cases with a very high proportion of cells classified as AML immature (median 86%, range 65-96%) whereas this proportion was much lower for the cases in cluster C3b (median 31%, range 1-89%; Fig. 4a). Instead, the cases in cluster C3b had a higher proportion of differentiated AML cells (classified as GMP and monocytes), but also a higher proportion of T cells (median 10%, range: 2-15% vs median 2%, range: 1-5% for cluster C3a; Supplementary Fig. 11b, c). We designated these two putative subtypes *NPM1* class I (Cluster C3a; *NPM1*class I) and *NPM1* class II (Cluster C3b; *NPM1*class II).

In order to study potential differences associated with the two gene expression profiles of *NPM1*-mutated AML in larger cohorts, we identified a robust set of genes (n = 180, false discovery rate <0.05) that were differentially expressed between the AML immature cells of *NPM1*class I and *NPM1*class II (Supplementary Data 3). To minimize the impact of the variable cell contents observed in bulk RNA samples, genes with notable expression in other cell types than AML immature were excluded. This list of 180 genes clearly distinguished between *NPM1*class I and *NPM1*class II also using bulk gene expression data from our local data set, classifying all 33 samples into either *NPM1*class I (n = 19) or *NPM1*class II (n = 14; Supplementary Fig. 12a, b, Supplementary Data 4). In addition, when applying this gene list on three external datasets of bulk RNA-seq data, from the TCGA[9], Beat-AML[34], and Clinseq[35] studies, we validated these findings by identifying 107 additional samples exhibiting the *NPM1*class I gene expression pattern (10 from TCGA, 46 from Beat-AML, and 51 from Clinseq) and 124 additional samples exhibiting the *NPM1*class II gene expression pattern (18 from TCGA, 74 from Beat-AML, and 32 from Clinseq; Supplementary Fig. 12c–h; Supplementary Data 5). A total of 116 samples from the three external cohorts, however, could not be reliably assigned to one of the two *NPM1* subtypes. These unclassifiable cases were found to contain a significantly smaller fraction of blasts in PB regardless of which sample tissue was used for RNA sequencing (Supplementary Fig. 12d, f, h), and were more likely to have a mature FAB subtype (M4 or M5; Supplementary Fig. 12d, f, h). We speculate that these cases (representing 33% of *NPM1*-mutated cases) may contain too few AML immature cells for classification into *NPM1*class I or *NPM1*class II based on bulk RNA-seq data and would require scRNA-seq to be accurately classified.

In total, the three external datasets expanded the total number of samples to 126 matching the *NPM1*class I expression profile and 138 matching the *NPM1*class II expression profile. In this expanded set, the two subtypes were found to be associated with different mutation

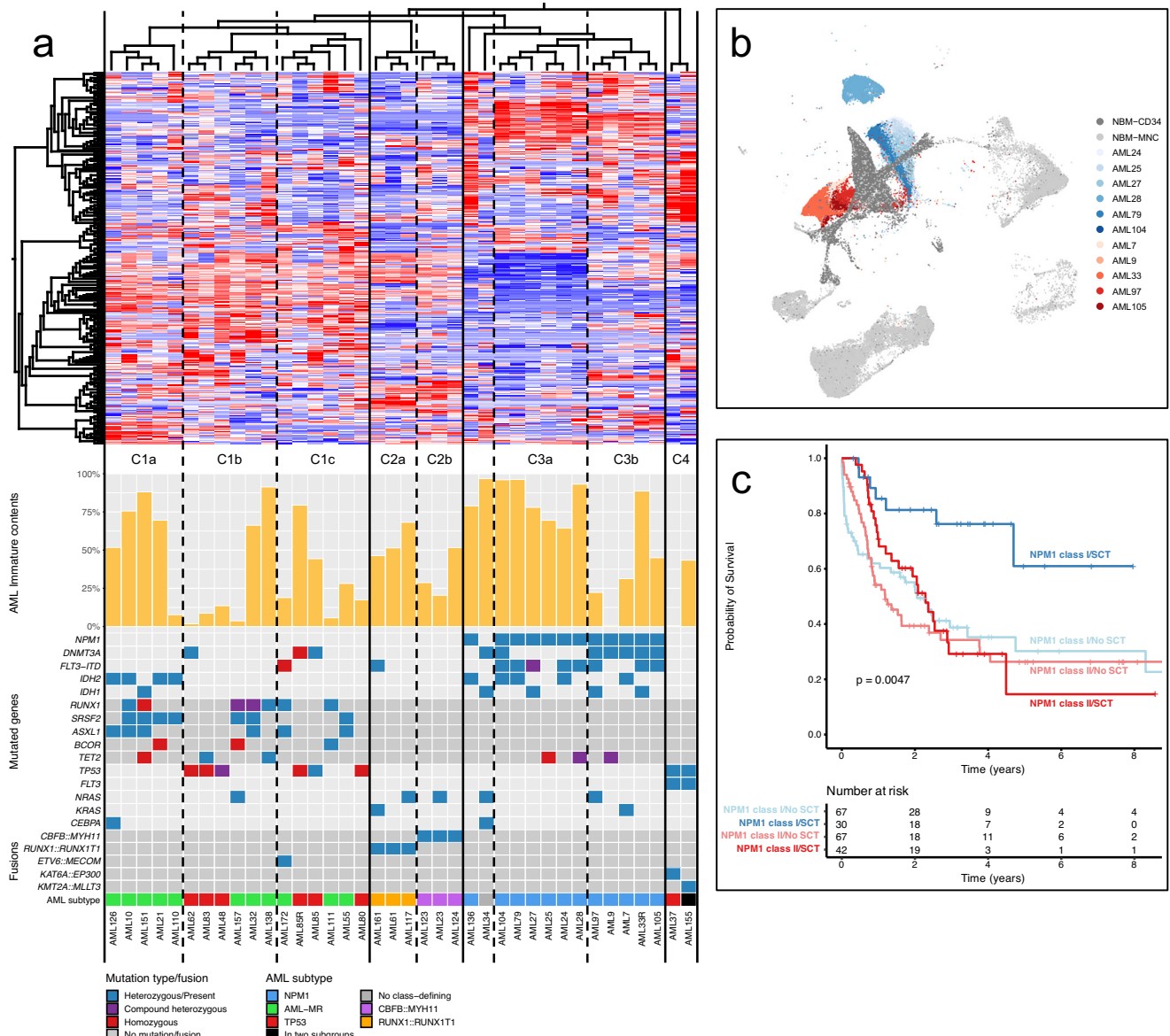

**Fig. 4 | Gene expression and survival features of *NPM1* class I and II.**
**a** Hierarchical clustering analysis based on the average gene expression profile of AML immature cells. The clustering and heatmap are based on the 361 most variable genes (threshold set at 0.365 of the highest standard deviation) between the AML immature expression profiles from 38 AML samples. The proportion of AML immature cells for each sample is indicated below the heatmap together with the most common mutations, fusion genes and the genetic subtype of each sample. Lines denote separated clusters (C1-C4) and dashed lines indicate subclusters (denoted a, b, or c), based on the dendrogram above the heatmap. **b** UMAP representation of 48,656 cells from eight normal bone marrow (NBM) samples (gray) and 28,937 AML immature cells from eleven *NPM1*-mutated AML samples. *NPM1*-mutated samples from subcluster C3a (*NPM1* class I) are indicated in different

shades of blue while *NPM1*-mutated samples from subcluster C3b (*NPM1* class II) are indicated in different shades of red. **c** Overall survival for *NPM1* class I (blue) and *NPM1* class II (red) for patients with hematopoietic stem cell transplantation (SCT, dark color) and without hematopoietic stem cell transplantation (No SCT, light color). Includes classifiable AMLs from TCGA, Beat-AML 2.0, and ClinSeq together with the current cohort (*n* = 206, two-sided Logrank test, *P* = 0.0047). The survival for *NPM1* class I with SCT (*n* = 30) is significantly better than *NPM1* class I without SCT (*n* = 67, two-sided Logrank test adjusted for multiple testing, *P* = 0.0033), whereas the survival for *NPM1* class II with SCT is not significantly different from *NPM1* class II without SCT (two-sided Logrank test adjusted for multiple testing, *P* = 0.43). Source data are provided as a Source Data file.

profiles. Mutations in *IDH2*, *TET2*, and *SRSF2* were significantly associated with *NPM1*[class I], whereas *FLT3*, *DNMT3A*, *WT1*, *NRAS*, and *PTPN11* mutations were significantly more common in the *NPM1*[class II] subtype (Supplementary Fig. 13a). Looking at the co-mutational patterns among the five most commonly mutated genes (*FLT3*, *DNMT3A*, *TET2*, *IDH2*, and *IDH1*), the most common patterns in *NPM1*[class I] cases were *FLT3* mutations together with either *TET2* or *IDH2* mutations, which were present in 35% of the cases in total (Supplementary Fig. 13b). The most common combination in *NPM1*[class II] cases was a *FLT3* and *DNMT3A* mutation occurring together. Mutations in *FLT3* and *DNMT3A* occurred

together in a total 42% of the cases, with or without additional mutations (Supplementary Fig. 13c).

To determine if this *NPM1* classification was associated with clinical outcome, we performed survival analysis on the four datasets, divided into discovery and validation cohorts (Fig. 4c, Supplementary Fig. 14). Notably, classification into *NPM1*[class I] and *NPM1*[class II] dictated the response to HSCT. While *NPM1*[class I] patients gained a substantial survival advantage from HSCT, *NPM1*[class II] patients did not seem to gain any survival advantage in these cohorts (Fig. 4c, Supplementary Fig. 14). This finding was consistent in all datasets and could not be

attributed to other factors beside the *NPM1*^class I and *NPM1*^class II classification.

Given the potential clinical application of this *NPM1* classification, we examined if a shorter gene list for distinguishing the classes could be identified, as this could enable a custom gene expression assay for clinical implementation. Hierarchical clustering based on a list of 30 genes, encompassing the ten most overexpressed genes in each of *NPM1*^class I, *NPM1*^class II and the unclassified cases, produced three distinct clusters in all datasets (Supplementary Fig. 15; Supplementary Data 6). In total, this method identified the correct *NPM1* class in 315/370 cases (85%). Most misclassifications occurred to or from the unclassified group (49/55 discordant cases), suspected to encompass cases with too few AML immature cells for classification based on bulk gene expression data. Thus, this indicates that it is possible to develop a custom assay for clinical identification of *NPM1* class, but that this should ideally involve a dedicated effort, including scRNA-seq data from a larger cohort, to also be able to characterize the unclassified group.

Another subdivision within *NPM1*-mutated AML was recently suggested based on bulk gene expression data, with two subgroups termed primitive and committed[36]. This gene expression classifier grouped the samples differently than the *NPM1*^class I and *NPM1*^class II subtypes (Supplementary Fig. 16a–f). In fact, the gene sets associated with the primitive and committed subgroups showed cell-specific gene expression patterns, indicating that this subdivision could reflect differences in cellular composition (Supplementary Fig. 17a, b). The *NPM1*^class I and *NPM1*^class II gene sets, however, displayed expression patterns consistent with intrinsic differences specifically within the AML immature cells (Supplementary Fig. 17c, d). In line with this, the two diagnose-relapse pairs in our cohort were noted to switch their committed/primitive classification between diagnosis and relapse, concordant with a classification that recognizes the cellular composition of the sample, which could differ between diagnosis and relapse. Similar switching was also noted for four of the thirteen patients with repeated samples in the Beat-AML cohort (Supplementary Fig. 16a, e). However, no instances of switching between *NPM1*^class I and *NPM1*^class II were observed in any of the cohorts, indicating that these subtypes are associated with intrinsic features within the immature AML cells that are stable over time.

In summary, we identified four gene expression clusters (C1-C4) based on gene expression differences in immature AML cells, as determined by scRNA-seq. A significant fraction (94%) of AML-MR and *TP53*-mutated AMLs clustered together, indicating that these subtypes may have similar molecular characteristics despite having distinct mutations. Notably, we identified two subtypes of *NPM1*-mutated AML, *NPM1*^class I and *NPM1*^class II. The first, *NPM1*^class I, had a homogenous cellular composition with mostly immature AML blast cells and improved survival in response to HSCT. In contrast, *NPM1*^class II had a heterogeneous cellular composition with a low proportion of immature AML blast cells and higher levels of monocytes and T cells, and showed no clinical benefit from HSCT.

### *NPM1* class I and *NPM1* class II have distinct MHC class II expression and sensitivity to allogeneic T cells

Analyzing the difference between the scRNA-seq expression profiles from *NPM1*^class I and *NPM1*^class II using gene set enrichment analysis (GSEA) revealed that AML immature cells from *NPM1*^class I had lower expression of genes involved in interferon signaling, interactions between lymphoid and non-lymphoid cells, and neutrophil degranulation, but higher expression of genes related to the DNA unwinding process that initiates DNA replication (Supplementary Fig. 18). A comparison with the other AML subtypes and HSCs from NBM showed that this difference, for the majority of gene sets, were due to downregulation in *NPM1*^class I samples since the expression level in *NPM1*^class II was similar to other AML subtypes and NBM HSCs (Supplementary

Fig. 19). Notably, several of the significantly downregulated gene sets in *NPM1*^class I included genes encoding major histocompatibility complex class II (MHC-II) molecules (i.e. *HLA-DM*, *-DO*, *-DP*, *-DQ*, and *-DR* genes). To further characterize this difference, we specifically studied the expression of the human leukocyte antigen (HLA) genes encoding MHC-II components, together with genes encoding MHC-I components, non-classical HLA genes, and the MHC-II master regulator CIITA[37] in AML immature cells from the scRNA-seq data (Fig. 5a,b and Supplementary Fig. 20). This confirmed a downregulation of all these genes in *NPM1*^class I samples when compared with *NPM1*^class II samples, other AML subtypes, and HSCs from NBM samples (Fig. 5a,b and Supplementary Fig. 20). The strongest downregulation was, however, observed for genes encoding MHC-II components and the regulator CIITA (Fig. 5a,b and Supplementary Fig. 20). Flow cytometry confirmed that MHC-II protein expression on the AML immature cells of the *NPM1*^class I subtype was significantly lower than that of both *NPM1*^class II AML immature cells and NBM HSCs (Fig. 5c, Supplementary Figs. 21, 22).

Since the most notable genetic difference between *NPM1*^class I and *NPM1*^class II was the presence or absence of a mutation in *DNMT3A*, we examined if higher expression of HLA genes encoding MHC-II components was a general feature associated with *DNMT3A*-mutated AML in the scRNA-seq data. Among *TP53*-mutated AML, AML immature cells from cases with concurrent *DNMT3A* mutations instead had lower MHC-II expression compared to those without *DNMT3A* mutation (Supplementary Fig. 23a, b). In addition, AML immature cells from the single *NPM1*^class I case harboring a *DNMT3A* mutation did not exhibit higher expression of HLA genes encoding MHC-II compared to other *NPM1*^class I cases (Supplementary Fig. 23c, d). Thus, the available evidence does not indicate a direct causal relationship between *DNMT3A* mutations and the higher MHC-II expression in *NPM1*^class II samples.

Based on the difference in survival following HSCT between *NPM1*^class I and *NPM1*^class II, we investigated the ex vivo sensitivity to allogeneic T cells using a coculture of AML cells and donor T cells. In this system, the *NPM1*^class II AML cells were significantly less sensitive to allogeneic T cells (mean 55% remaining cells after treatment vs 23% for *NPM1*^class I; $P = 0.014$; Fig. 5d), indicating that the difference in survival between *NPM1*^class I and *NPM1*^class II following allogeneic HSCT could be related to different capabilities of the AML cells to suppress allogeneic T cells.

MHC-II molecule expression can be induced by interferon-gamma (IFNg) through the intermediate regulator CIITA[37]. We therefore examined the effect of IFNg treatment on AML cells from *NPM1*^class I (with low MHC-II expression) and *NPM1*^class II (with high MHC-II expression; Supplementary Fig. 24). This showed that IFNg induced MHC-II upregulation in both *NPM1* subtypes, and was associated with an increase in T cell sensitivity in the majority of samples from both subtypes (Supplementary Fig. 24), highlighting that T cell sensitivity in *NPM1*^class I can be further increased by upregulating MHC-II, but also that the suppression of allogeneic T cells by *NPM1*^class II AML cells occurs independently of MHC-II expression.

To examine if the difference in T cell response between *NPM1*^class I and *NPM1*^class II instead could be mediated through checkpoint receptor signaling[38], we analyzed the expression of checkpoint molecules on the surface of AML cells (PD-L1, PD-L2, VISTA, CD80, CD86, CD155, Galectin-9, CD47, and CD200) and T cells (CTLA4, LAG3, PD1, TIGIT, and TIM3) in diagnostic *NPM1*^class I and *NPM1*^class II BM samples using flow cytometry (Supplementary Fig. 25 and Supplementary Fig. 26). The expression of immune checkpoint ligands was found to be similar between the two subtypes in both immature and mature cell compartments, but with potentially higher expression of CD86 and VISTA in *NPM1*^class II (Supplementary Fig. 25). *NPM1*^class II samples also exhibited higher expression of inhibitory receptors such as TIM3 and TIGIT in certain T cell populations (Supplementary Fig. 26). These differences

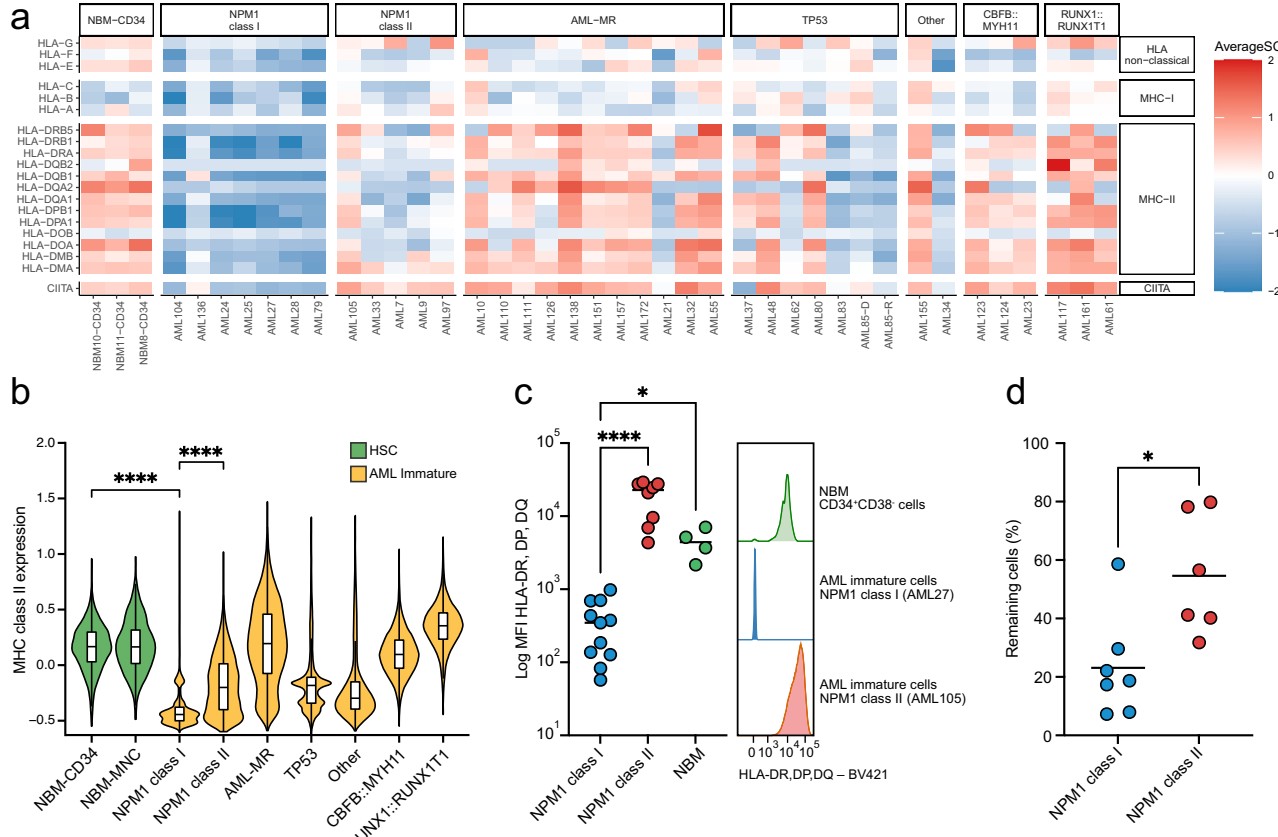

**Fig. 5 | Immune evasion in *NPM1* class I and II. a** Average expression of genes encoding MHC class I (MHC-I) and II (MHC-II) components, non-classical HLA genes, and the MHC-II master regulator CIITA, based on 98,022 AML immature cells from 38 AML samples and 5958 hematopoietic stem cells (HSC) from three normal bone marrow (NBM) samples. **b** Expression of MHC class II genes in HSC cells from NBM-CD34 ($n$ = 5958 cells) and NBM-MNC ($n$ = 282 cells) and in AML immature cells from the subtypes *NPM1* class I ($n$ = 21,458 cells), *NPM1* class II (n = 10,382 cells), AML-MR ($n$ = 26,549 cells), *TP53* ($n$ = 12,142 cells), Other ($n$ = 9565 cells), *CBFB::MYH11* ($n$ = 8026 cells), and *RUNX1::RUNX1T1* ($n$ = 9900 cells), based on single cell RNA-sequencing data. Box-and-whisker plots inside violin plots show median (center line), first and third quartiles (hinges), and 1.5x interquartile range (whiskers). The expression was significantly lower in AML immature cells from *NPM1* class I samples compared to both HSC cells from NBM-CD34 (two-sided Mann-Whitney U test, $P$ < 2.2e−16) and AML immature cells from *NPM1* class II samples

(two-sided Mann-Whitney U test, $P$ < 2.2e−16). **c** Expression of HLA-DR, HLA-DP and HLA-DQ on the cell surface of CD34 + CD38− cells from NBM and AML immature cells from *NPM1*-mutated samples by flow cytometry. Gating of AML immature cells was based on markers determined by single-cell antibody-derived tag-sequencing (gating strategy described in Supplementary Figs. 21, 22). Left panel: the geometric mean fluorescence intensity (MFI) values were significantly lower for *NPM1* class I samples ($n$ = 11 samples) compared to both *NPM1* class II ($n$ = 8 samples; two-sided Mann-Whitney U test; $P$ = 0.00003) and NBM ($n$ = 4 samples; two-sided Mann-Whitney U test; $P$ = 0.0372). Right panel: histograms for AML immature cells from representative samples for each group and CD34 + CD38− cells from NBM.
**d** Survival of AML cells after three-day ex vivo coculture with allogeneic T cells. *NPM1* class II AML cells ($n$ = 6 samples) were significantly less sensitive to allogeneic T cells than *NPM1* class I AML cells ($n$ = 7 samples, two-sided Mann-Whitney U test, $P$ = 0.014). Source data are provided as a Source Data file.

could potentially contribute to the resistance of *NPM1*^class II AML cells to T cell mediated killing observed both ex vivo and clinically in the HSCT setting. In keeping with these findings, a low but significant increase in myeloid cells with surface expression of Galectin-9 was also observed in the *NPM1*^class II subtype (Supplementary Fig. 25), suggesting a potential T cell suppression mechanism through interaction between Galectin-9 and TIM3 on T cells[39]. Two of four *NPM1*^class II samples also harbored T cells expressing both PD1 and TIM3 (Supplementary Fig. 27), indicating that the TIM3 expression may be coupled to T cell exhaustion[40]. This was not observed in any of the four *NPM1*^class I samples. In addition, we compared the distribution of T cell subsets within the CD4^+ and CD8^+ compartments, based on scRNA-seq cell classifications and flow cytometry of diagnostic AML samples (Supplementary Fig. 28). This showed a trend towards a higher proportion of CD8 naive cells in both scRNA-seq and flow cytometry data ($P$ = 0.051 and $P$ = 0.065), suggesting microenvironmental differences between the two *NPM1* classes affecting the T cell composition already at diagnosis.

We conclude that *NPM1*^class I and *NPM1*^class II AML employ different strategies for immune evasion. *NPM1*^class I AML is associated with MHC-II downregulation, while *NPM1*^class II AML displays a distinct resistance to T cell killing. The difference in immune evasion strategies and the resistance of *NPM1*^class II AML cells to allogeneic T cells could explain the difference in survival between the two *NPM1* subtypes following HSCT, where *NPM1*^class I responds well whereas *NPM1*^class II AML lacks clinical benefit.

## Discussion

Despite improvements in the genomic classification of AML, this disease is associated with substantial heterogeneity in treatment response and clinical outcome. Here, we used single-cell transcriptomic data to dissect the transcriptional profiles of bulk samples, defining the transcriptional and cellular state space of AML. This unveiled a marked cellular heterogeneity of genetic AML subtypes and revealed two distinct transcriptomic subtypes of *NPM1*-mutated AML with different immune evasion properties and response to

hematopoietic stem cell transplantation; results with direct clinical implications for the treatment of patients with AML.

Bulk RNA-seq revealed striking gene signatures enriched for cell type markers, indicating that the bulk expression profiles of AML samples were severely affected by the cellular composition of the samples. The concept of utilizing cell signatures for deconvolution of bulk gene expression data for AML classification has been explored in several recent studies[34,41,42], for example, showing the impact of cellular composition in AML on treatment response[42]. While this approach can provide a view of the cellular constitution of an AML sample, the gene expression profiles within the different cell types cannot be defined. Thus, comparing the gene expression between samples for a defined cell type or studying the effect of mutations in different cell types is not feasible using bulk RNA-seq. In addition, more complete AML single cell data sets might be needed for reliable deconvolution of bulk data since the single cell AML data sets published to date have been generated from mononuclear cells[25,27–30], while several of the mature cell types with the strongest impact on the bulk gene expression profile, as identified here, were enucleated (erythroid precursors and platelets) or multinucleated (neutrophils).

To determine the gene expression profiles of the constituent cell types in AML, we performed single-cell multimodal sequencing. This confirmed that there is a substantial heterogeneity with regard to cellular composition and maturation of mutated AML cells, in concordance with previous single-cell RNA-seq studies of AML cell composition[25–27]. In this larger and genetically characterized data set, however, we observed a high degree of cellular heterogeneity even among AMLs of the same genetic subtype. In addition, the distinct gene expression profiles of immature AML cells as determined by scRNA-seq were found to align better with the underlying genetic subtype than the bulk expression profiles for AMLs with *NPM1* mutations and the fusion genes *CBFB::MYH11* and *RUNX1::RUNX1T1*. However, AMLs from the AML-MR and the *TP53*-mutated subtype together formed the largest gene expression class, which was subdivided into smaller classes that did not follow the genetic subdivision. Thus, these two genetic subtypes, which are both associated with myelodysplastic syndrome and secondary AML developing from this disease[43,44], display shared gene expression profiles despite having minimal overlap in somatic mutations.

Notably, *NPM1*-mutated AMLs were subdivided into two gene expression classes, distinct from previously described subtypes among *NPM1*-mutated AMLs identified using bulk gene expression data[36]. The *NPM1*class II subtype was associated with a diverse cellular content with a high fraction of monocytes and T cells. However, the subtype-specific gene expression pattern was also present in at least one relapse sample with a low proportion of monocytes and T cells, indicating that the distinct gene expression profile is not a response to extrinsic pressures but an intrinsic feature of the immature AML cells of this subtype. In contrast with *NPM1*class II, the *NPM1*class I subtype was associated with a high proportion of immature AML cells that displayed downregulation of MHC class II components as a likely immune evasion strategy. Interestingly, it has previously been noted in flow cytometry-based studies of *NPM1* positive AML that a subset of cases has downregulated MHC class II cell surface expression[45,46], but that this is a distinct subtype with a specific global gene expression profile has not previously been described. Importantly, when comparing the overall survival between *NPM1*class I and *NPM1*class II following HSCT, we also observed a vastly superior outcome for *NPM1*class I, whereas *NPM1*class II cases did not seem to have any survival advantage following HSCT. In line with these observations, *NPM1*class II AML cells exhibited reduced sensitivity to allogeneic T cell-mediated cytotoxicity ex vivo, suggesting an intrinsic mechanism by which this *NPM1* subtype evades elimination by the immune system following HSCT. Comprehensive characterization of checkpoint molecule expression on both AML cells and T cells revealed no general differences between *NPM1*class I and

*NPM1*class II. However, checkpoint expression was notably more heterogeneous in the *NPM1*class II subgroup. Several *NPM1*class II cases displayed upregulation of individual receptors and ligands – a pattern not observed in *NPM1*class I. In addition, both scRNA-seq and flow cytometry analyses revealed a trend toward a higher proportion of CD8 naive cells in *NPM1*class II, suggesting differences in T cell functionality between the subtypes. Furthermore, co-expression of TIM3 and PD1 in two out of four *NPM1*class II cases indicated the presence of exhausted T cells. The most striking divergence between these subtypes is their post-HSCT survival. Our findings indicate that this disparity may be explained by differences in the intrinsic susceptibility of the immature AML cells to allogeneic T cell killing.

The two *NPM1* subtypes were associated with different mutational patterns, with co-occurring *DNMT3A* and *FLT3*-ITD mutations being the most common composition in *NPM1*class II. AML harboring co-occurring *NPM1*, *DNMT3A*, and *FLT3*-ITD mutations have previously been described to be associated with an adverse prognosis[10,47,48]. Notably, however, this genetic setup occurred in less than half of the *NPM1*class II cases and was also present in a small number of *NPM1*class I cases. Thus, while the inferior prognosis of AML with *NPM1*, *DNMT3A*, and *FLT3*-ITD is likely to be related to its over-representation in *NPM1*class II, this genotype in itself is not a suitable surrogate marker for *NPM1*class II.

In conclusion, we here describe how single-cell sequencing can be used to analyze the transcriptional complexity in AML, and to unravel the cellular heterogeneity that clouds bulk RNA sequencing. This methodology enabled the detection of two expression-based *NPM1* subtypes, *NPM1*class I and *NPM1*class II. The two classes display distinct modes of interaction with the immune system, with *NPM1*class I showing a downregulation of MHC-II as an immune evasion strategy, whereas *NPM1*class II exhibits enhanced resistance to the anti-leukemia T cell response. Our data further suggests that *NPM1*class I has a significant survival benefit from HSCT, whereas *NPM1*class II does not. This transcriptional subtype information is expected to have direct clinical implications for the treatment of patients with AML.

## Methods
### Patient samples and integrative sequencing
This research complies with all relevant ethical regulations. Primary samples were collected at Skåne University Hospital after written informed consent in accordance with the Declaration of Helsinki. The study was approved by the Ethics Committee of Lund University with reference numbers dnr 2011/289 and dnr 2023-01550-01. Bone marrow aspirates (BM), peripheral blood (PB), and skin biopsies were collected from AML participants. BM was also collected from healthy controls. AML samples were included following diagnosis of de novo AML, secondary AML, therapy-related AML, or relapse of previously diagnosed AML. In total, 120 AML samples were included, representing 112 individuals. Of these, 97 cases were represented by a single sample at diagnosis, eight cases were represented by paired diagnosis and relapse samples, one case was represented by a sample taken during the disease, and six cases were represented by a single sample taken at relapse. Donors of normal bone marrow received modest financial compensation. There was no other compensation for study participation. Primary leukemia samples were collected from both male and female patients to allow for equitable implementation of clinically relevant findings. Skin biopsies were collected from all 112 individuals; these were cultured to eliminate infiltrating AML cells and then used as normal reference material.

The following techniques were used to characterize the AML samples at the genomic level (Supplementary Data 1): whole exome sequencing (WES; paired tumor-normal analysis for all 120 samples), mate pair whole genome sequencing (MP-WGS; tumor only analysis for 98 samples), targeted PCR for *CEBPA* and *FLT3*-ITD with sequencing readout (paired tumor-normal analysis, 120 samples). WES libraries

were prepared from 50 ng DNA using the Nextera rapid exome kit (Illumina) according to the manufacturer's instructions (Nextera rapid capture enrichment guide). MP-WGS libraries were prepared from 1 μg of DNA using the Nextera mate pair library preparation kit (Illumina) according to the manufacturer's instructions (Nextera mate pair library prep reference guide). The targeted PCR for *CEBPA* and *FLT3*-ITD was performed in a multiplex reaction using the primers: AGG CACCGGAATCTCCTAGT (*CEBPA* forward), CCTGCCGGGTATAA AAGCTG (*CEBPA* reverse), AACTGTGCCTCCCATTTTTG (*FLT3*-ITD forward), CCTGATTGTCTGTGGGGAGT (*FLT3*-ITD reverse). PCR amplification was performed using the Q5 hot start high-fidelity DNA polymerase (New England Biolabs) with Q5 high GC enhancer included in the amplification mix. The annealing temperature was set to 64 °C. Sequencing libraries were then prepared from 1 ng of amplified material using the Nextera XT library preparation kit (Illumina) according to the manufacturer's instructions.

RNA-sequencing libraries were prepared from poly-A selected RNA using the TruSeq RNA library preparation kit v2 (Illumina) according to the manufacturer's instructions, but with a modified RNA fragmentation step lowering the incubation time at 94 °C from 8 minutes to 10 seconds to allow for longer RNA fragments, as previously described[14]. All libraries were sequenced using a NextSeq 500 (Illumina).

The 120 AML cases (representing 112 individuals) were, based on somatic variants (SNVs, indels, copy number changes, and fusion genes), classified into thirteen molecular AML subtypes as described by Papaemmaniul et al.[10]: *NPM1* (*n* = 33), myelodysplasia-related gene mutations (AML-MR; one or more driver mutations in *RUNX1*, *ASXL1*, *BCOR*, *STAG2*, *EZH2*, *SRSF2*, *SF3B1*, *U2AF1*, *ZRSR2*, or *MLL*[PTD], or, in the presence of other class-defining lesions, two or more mutations were required) (*n* = 30), *TP53* (and/or chromosomal aneuploidy; *n* = 18), *CBFB::MYH11* (n = 8), double mutated *CEBPA* (biallelic *CEBPA*; *n* = 1), *PML::RARA* (*n* = 8), *RUNX1::RUNX1T1* (*n* = 3), x::*KMT2A* (*KMT2A* fusion; *n* = 3), inv(3)(q21q26.2) or t(3;3)(q21;q26.2) (*GATA2::MECOM*; = 1), *IDH2*[R172] (and no other class-defining lesions) (*n* = 0), *DEK::NUP214* (*n* = 2), no class-defining lesions (no class-defining; n = 8), no detected driver (*n* = 3), and finally, meeting criteria for ≥2 genomic subgroups (In two subgroups; *n* = 2). The subtypes previously referred to as chromatin-spliceosome and *MLL* fusion[10] are here referred to as AML-MR and *KMT2A* fusion to align with updated nomenclature.

## Bulk sequencing data analysis

The sequencing reads from WES, MP-WGS, and targeted PCR were aligned against the human genome (hg19) using bwa[49] v0.7.15 (https://github.com/lh3/bwa).

Somatic (tumor-normal) single nucleotide variant and small indel calling in the WES and targeted PCR data was performed using Strelka v0.4.7[50], Strelka v2.9.4[51] (https://github.com/Illumina/strelka), Mutect2[52] (from gatk 4.0.8.1; https://github.com/broadinstitute/gatk), and freebayes[53] 1.1.0 (https://github.com/freebayes/freebayes). For freeabyes, a somatic score (SSC) was calculated from the provided genotype likelihoods (GL) for tumor (T) and normal (N), using the formula SSC = T.GL(T) - T.GL(N) + N.GL(N) - N.GL(T).

Larger structural variants in the WES and targeted PCR data were called using pindel[54] v0.2.5b8 (https://github.com/genome/pindel), and manta[55] v1.4.0 (https://github.com/Illumina/manta). All variants detected in WES were filtered using the reported quality score for each variant caller, (strelka v0.4.7 and v2.9.4 SNVs, QSS > = 100; strelka v0.4.7 indels, QSI > = 30; strelka v2.9.4 indels, QSI > = 100; Mutect2, TLOD > = 30; freebayes, SSC > = 60; manta, SOMATICSCORE > = 30; pindel, AD > = 10, VAF > 10%). More strict filtering criteria were used for variants in targeted PCR data for the following variant callers (strelka v0.4.7 and v2.9.4 SNVs, QSS > = 300; strelka v0.4.7 indels, QSI > = 60; strelka v2.9.4 indels, QSI > = 100; freebayes, SSC > = 300; same filter settings as before for Mutect2, manta, and pindel). Variants detected in

any of the cultured skin biopsies (not only the corresponding skin biopsy) were removed as suspected germline variants. Somatic variants were annotated using SnpEff v0.4.3r (https://pcingola.github.io/SnpEff/). Plots illustrating somatic data were produced using R v4.1.2 (https://www.R-project.org).

Copy number detection was performed on WES data using cnvkit[56] v0.9.2 (https://github.com/etal/cnvkit). Structural variants in the MP-WGS data were called using delly[57] v0.7.7 (https://github.com/dellytools/delly).

Gene expression values (fragments per kilobase of transcript per million reads; fpkm) were calculated from RNA-seq reads using RSEM v1.2.30[58] (https://github.com/deweylab/RSEM) and fusion genes were called using chimerascan[59] v0.4.5a (https://code.google.com/archive/p/chimerascan/). Only fusion genes detected both in MP-WGS data and RNA-seq data, or those that were previously described in AML, were included.

Hierarchical clustering was performed using Qlucore Omics Explorer v3.7 (Qlucore). Gene sets for identifying the *NPM1* subtypes described by Mer et al.[36] were retrieved from the supplementary data from their publication.

## External datasets

Somatic SNV and indel variants described in the TCGA AML data set[9] were retrieved from the plain text (tsv) version of their supplemental table 6, available at https://gdc.cancer.gov/about-data/publications/laml_2012. Gene expression data from the TCGA AML data set were downloaded as RSEM normalized counts from cbioportal (https://www.cbioportal.org/datasets). Somatic SNV and indel variant data from the Beat-AML 1.0 data set[31] were exported from http://www.vizome.org/aml/geneset/. Gene expression data from the Beat-AML 1.0 data set were downloaded as RSEM normalized counts from cbioportal (https://www.cbioportal.org/datasets). Gene expression data from the Beat-AML 2.0 data set[34] were downloaded as normalized expression data from https://biodev.github.io/BeatAML2/. Somatic SNV and indel variant data from the data set published by Papaemmanuil et al.[10] were downloaded from https://github.com/gerstung-lab/AML-multistage/tree/master/data. Cell type markers from the panglaoDB single cell database were downloaded from https://panglaodb.se/markers/PanglaoDB_markers_27_Mar_2020.tsv.gz. Gene expression data and somatic SNV and indel variant data from 95 AMLs with *NPM1* mutations from the Clinseq[35] dataset were provided directly by the authors, together with data from 32 additional AMLs.

## Single-cell RNA sequencing

Single-cell RNA sequencing (scRNA-seq) was performed on 38 AML samples from the subtypes *NPM1* (*n* = 12), AML-MR (*n* = 11), *TP53* (*n* = 7), *CBFB::MYH11* (*n* = 3), *RUNX1::RUNX1T1* (*n* = 3), AML without class-defining mutations (*n* = 1), and AML meeting the criteria for two subtypes (n = 1). In addition, NBM samples with mononuclear cells (*n* = 5), and CD34 + -enriched cells (*n* = 3) were included. Of these, 34 AML samples and six NBM samples were also subjected to antibody derived tag sequencing using a panel of 10-14 AML stem cell markers. Cryopreserved mononuclear cells were gently thawed and stained with TotalseqA antibodies (BioLegend) and draq7 (Biostatus), according to each manufacturer's protocol. The cell suspensions were sorted to contain viable single cells, based on DRAQ7 staining, using an Aria Fusion (BD Biosciences). The scRNA-seq and ADT-seq libraries were prepared using Chromium Single Cell 3′ Reagent Kits v3 according to the manufacturer's instructions (10X Genomics) and sequenced on a Novaseq 6000 System (Illumina).

## Single-cell RNA sequencing data analysis

Raw sequencing data were converted to fastq format using bcl2fastq (Illumina). The reads were aligned to the human reference genome (hg19/GRCh37) and converted to single-cell transcript and ADT count

matrices using cellranger count (10x Genomics, v.3.1.0). Further analysis of scRNA-seq and scADT data was performed using Seurat (v.4.0.0)[60]. Low-quality cells, with less than 200 informative genes or more than 15% of the detected transcripts derived from mitochondrial genes, were excluded. Data from all samples (in total 245,073 cells) were merged into a single object using Seurat's merge function. Normalization was performed using sctransform (v.0.3.2)[61] for transcript data and using centered log ratio for scADT data. For visualization and clustering, the dimensionality of the normalized scRNA data was further reduced using principal component analysis (PCA) with 80 principal components. The data was visualized using uniform manifold approximation and projection (UMAP)[62]. Cell type prediction was performed first on cells only from NBM samples using a well-defined external reference dataset of NBM cells[63] using Seurat's TransferData function. The predicted cell types for the local NBM samples were then used as a reference for cell type prediction for the remaining cells using Seurat's TransferData function. The cells were clustered using shared nearest neighbor modularity optimization-based clustering[64] with resolution 0.5. Cells in clusters that consisted of more than 80% cells from AML samples and where a majority of the cells were predicted to be HSC or LMPP according to the NBM reference were denoted AML immature cells instead of their initial predicted cell type. Visualization of NBM reference knn force graphs[65] and projection of AML single cell data onto these graphs were performed using SingleCellProjections.jl (https://github.com/BioJulia/SingleCellProjections.jl). For hierarchical clustering of AML immature gene expression data, the average expression based on cell type and AML sample was first calculated using Seurat's AverageExpression function. Log2 transformation (with a threshold at 0.01) and hierarchical clustering of this data was performed using Qlucore Omics Explorer v3.7 (Qlucore).

## NPM1 class I and II expression profiles

The *NPM1* class I and II specific expression profiles were identified using Qlucore Omics Explorer v3.7 (Qlucore) by first selecting the most significantly upregulated genes in immature cells compared to other cell types in the *NPM1* class I and II samples. For *NPM1* class I, the 548 most highly expressed genes in immature cells were identified by performing a t-test between the average expression in immature cell types (AML immature, GMP, and LMPP) compared to mature cell types (Monocytes, Erythroid cells, Dendritic cells, NK cells, T cells, and B cells) in *NPM1* class I samples, and then filtered based on q-values calculated using the Benjamini-Hochberg method ($q < 0.0000225$). For *NPM1* class II, the 389 most highly expressed genes in immature cells were identified by the same procedure for *NPM1* class II samples ($q < 0.00003$). Once these two gene sets (i.e., genes upregulated in immature AML cells compared to mature cell types) had been identified (containing a total of 908 genes), a t-test was performed to identify the most significantly differentially expressed genes from these two gene sets between AML immature cells in *NPM1* class I and class II samples. After filtering at $q < 0.05$, this resulted in a list of 180 genes (Supplementary Data 3). Hierarchical clustering based on this gene list was applied to bulk RNA-seq data from the local and external cohorts to identify samples with *NPM1* class I and class II expression profiles.

## NPM1 class I/NPM1 class II gene set enrichment analysis

Gene set enrichment analysis for the difference between the average gene expression in *NPM1* class I compared to class II was performed using a gene list ranked by fold change in average expression between *NPM1* class I AML immature cells and *NPM1* class II AML immature cells, produced using Qlucore Omics Explorer 3.7 (Qlucore). Gene set enrichment analysis[66] was performed using clusterprofiler[67] based on the Reactome pathway database[68] curated gene set from msigdb[69] (v7.5.1).

## Single cell RNA-seq mutation calling

Single-cell RNA mutation sequencing (scRNAmut-seq) was performed by targeted PCR amplification of known somatic mutations from leftover full-length amplified cDNA from the 10X genomics library preparations (Supplementary Fig. 4). PCR primers were designed against all variants found in WES data for each patient using primer3[70] v2.5.0 (https://github.com/primer3-org/primer3). Primers were then evaluated based on transcript coverage in scRNA and bulk RNA-seq data and the distance between the mutation and the 3′ end of the gene. Primers for detecting 1-4 somatic mutations were selected for each sample with scRNA-seq data. All primers were produced as custom ultramer DNA oligos (Integrated DNA technologies). Prior to scRNAmut-seq the full-length amplified cDNA material was repaired using PreCR repair mix (New England Biolabs) according to the manufacturer's instructions. The preparation of scRNAmut-seq libraries involved two PCR reactions (Supplementary Fig. 4). The first PCR was a targeted PCR using a mutation-specific primer (primer sequences listed in Supplementary Data 2) with an overhang of a partial read 2 (right primer) and read 1 sequencing primer (left primer). The reaction was performed using KAPA HiFi HotStart ReadyMix (Roche) in 50 μl reaction volume with the following protocol: initial denaturation at 95 °C, 180 s; 10 cycles of denaturation at 98 °C, 20 s; annealing at 60 °C, 15 s; extension at 72 °C, 60 s; and lastly final extension at 72 °C, 300 s. The second PCR was a sample index PCR with primers adding the P5 sequence (left primer) as well as i7 index and P7 sequences (right primer) needed for Illumina bridge amplification (Supplementary Fig. 4). The primers used in the second PCR were: left primer: 5′-AATGATACGGCGACCACCGAGATCTACACTCTTTCCCTACACGACGCTC-3′, and right primer: 5′-CAAGCAGAAGACGGCATACGAGATNNNNNNNNGTCTCGTGGGCTCGG-3′. The following i7 index sequences were used (in place of the 8 N stretch in the reverse primer): TCGCCTTA, CTAGTACG, TTCTGCCT, GCTCAGGA, AGGAGTCC, CATGCCTA. A 25 ul reaction was prepared with PCR-grade water, KAPA HiFi HotStart ReadyMix (Roche), primers, and DNA template from the targeted PCR reaction. The following PCR protocol was used for the second PCR: initial denaturation at 95 °C, 180 s; 10 cycles of denaturation at 98 °C, 20 s; annealing at 54 °C, 30 s; extension at 72 °C, 60 s; and lastly final extension at 72 °C, 300 s. The scRNAmut-seq libraries were purified using AMPure XP beads (Thermo Fisher Scientific) prior to sequencing on a Novaseq 6000 System (Illumina).

## Single-cell RNA-seq mutation calling data analysis

Raw sequencing data were demultiplexed and converted to fastq format using bcl2fastq (Illumina). For variant calling of individual reads, a diploid reference genome based on hg19 but including the known targeted somatic variants of each case was constructed using STAR[71] v2.7.8a (https://github.com/alexdobin/STAR). The reads were then aligned to this reference genome using STAR, with command line options to include haplotype information (i.e., if the read matches reference or somatic variant) together with the 10X UMI and cellular barcode information. Reads covering the targeted mutations matching either the reference or the somatic variant were counted for each combination of cell barcode and UMI. The genotype for each cell barcode-UMI combination was classified as mutated or reference based on the most common genotype among the reads. To infer the proportion of mutated cells in a group of cells (of the same cell type or projected within the same hexagon), the genotype classification of UMI:s covering somatic heterozygous mutations only, within any of the cells in the group, was considered. The proportion of mutated cells ($P$) was estimated by dividing the number of UMI:s classified as mutated ($n_m$) by the total number of UMI:s (with mutated or reference genotype) from that site ($n_m + n_r$), which was then divided by

0.5 (i.e., the expected allele fraction if all cells were mutated):

$$P = \frac{n_m}{n_m + n_r} / 0.5 \qquad (1)$$

### Flow cytometry analysis of MHC class II expression

AML and NBM samples were thawed in complete medium with DNase and then washed in PBS 2% FBS. A total amount of $0.2–1 \times 10^6$ cells was used for flow cytometry, while the remaining cells were put in culture. Staining was performed for 20 minutes at 4 °C in a 100 μl mix of appropriate antibodies in PBS 2% FBS, including CD33–PE (1:100), CD117–AF488 (1:100), C3AR–PE/Cy7 (5:100), GPR56–APC (1:100), CD123–BV711 (1:100) and HLA-DR, DP, DQ–BV421 (1:100) for AML sample analysis; and CD3–PE/Cy7 (1:100), CD19–APC/Cy7 (3:100), CD34–AF488 (3:100), CD38–APC/Cy7 (3:100) and HLA-DR, DP, DQ–BV421 (1:100) for NBM analysis. After staining, cells were washed with PBS 2% FBS and resuspended in 200 μl PBS 2% FBS with 1:100 7–AAD for viability staining. Samples were analyzed using an LSR Fortessa flow cytometer (BD Biosciences), and the data were analyzed using FlowJo 10 (BD Biosciences). Identification of the AML immature populations in AML samples was made with the help of the scADT-seq data, when possible (Supplementary Figs. 21, 22). If no scADT-seq data were available, AML immature was defined as CD33+CD117+.

### AML sample culture

AML cells remaining after flow cytometry analysis were put in culture to evaluate the response to Interferon gamma (IFNg). The AML cells were cultured in Iscove's modified Dulbecco's medium (IMDM) supplemented with 1% penicillin/streptomycin (P/S), 1% L-glutamine, 0.1 mM b-mercaptoethanol, 15% bovine serum albumin, insulin, transferrin (BIT 9500; StemCell Technologies), SCF (100 ng/mL; all cytokines from Peprotech), FLT3-L (50 ng/mL), IL3 (IL3; 20 ng/mL), gCSF (20 ng/mL), StemRegenin-1 (500 nM; StemCell Technologies) and UM729 (500 nM; StemCell Technologies). IFNg was added to appropriate wells at a final concentration of 50 ng/mL. The cells were cultured for 3 days at 37 °C, 5% $CO_2$, and MHC class II expression was reassessed by flow cytometry after IFNg treatment as before. The cells were then washed in PBS 2% FCS, counted, and replated accordingly for the T cell assay.

### T cell co-culture

For the T cell assay, CD3+ T cells were isolated from PB-MNCs using MACS cell separation columns with the CD3 microbeads isolation kit, according to the manufacturer's instructions (Miltenyi Biotec). The AML cells were cultured in Iscove's modified Dulbecco's medium (IMDM) supplemented with 1% penicillin/streptomycin (P/S), 1% L-glutamine, 0.1 mM b-mercaptoethanol, 15% bovine serum albumin, insulin, transferrin (BIT 9500; StemCell Technologies), SCF (100 ng/mL; all cytokines from PeproTech), FLT3-L (50 ng/mL), IL3 (20 ng/mL), gCSF (20 ng/mL), StemRegenin-1 (500 nM; StemCell Technologies), and UM729 (500 nM; StemCell Technologies). After culturing for 3 days, AML cells were washed and counted, and an equal number of cells were plated in 100 μl RPMI medium (supplemented with 1% P/S and 10% FBS) per well in triplicate or quadruplicate ($5–10 \times 10^3$ cells/well). Then, equal numbers of T cells were added in an additional volume of 100 μl RPMI per well, to a final volume of 200 μl and a ratio of 1:1 target to effector cell. IL2 was added to a final concentration of 10 ng/mL, and the mixed culture was cultured for an additional 3 days at 37 °C, 5% $CO_2$. After 3 days, cell numbers were determined by flow cytometry using CountBright beads (Life Technologies) and staining with CD3–PE/Cy7 and CD33–BV421.

### Flow cytometry analysis of immune-related markers on AML samples

AML samples were thawed in complete medium with DNase and then washed in PBS 2% FBS. Each sample was divided for analysis with corresponding panels for AML and T cells, and staining was performed for 20 minutes at 4 °C in a 100 μl mix of appropriate antibodies in PBS 2% FBS. The panels included combinations of the following antibodies:

–AML cell analysis: CD33–APC/Cy7 (1.5:100), CD3–BV510 (1.5:100), CD19–BV510 (1.5:100), CD14–APC (1:100), CD123–BV711 (1:100), PDL2–BV421 (5:100), PDL1–FITC (5:100), CD200–PE/Cy7 (5:100), CD155–BV421 (5:100), VISTA–PE (5:100), GAL9–PE/Cy7 (5:100), CD47–BV421 (5:100), CD80–FITC (5:100), CD86–PE (1:100).

–T cell analysis: CD3–PE (1.5:100), CD4–BV711 (1:100), CD8–APC/Cy7 (1:100), CD45RA–BV510 (3:100), CCR7–PE/Cy7 (5:100), PD1–BV421 (3:100), TIM3–FITC (3:100), LAG3–APC (3:100), CTLA4–BV421 (3:100), TIGIT–APC (3:100).

After staining, cells were washed with PBS 2% FBS and resuspended in 200 μl PBS 2% FBS with 1:100 7–AAD for viability staining. Samples were analyzed using an LSR Fortessa flow cytometer (BD Biosciences), and the data were analyzed using FlowJo 10 (BD Biosciences). Within the CD4+ and CD8+ T cell populations, naïve cells were defined as CCR7+CD45RA+, effector cells as CCR7-CD45RA+, effector memory cells as CCR7-CD45RA-, and central memory cells as CCR7+CD45RA-. Within the CD33+ AML population, immature AML cells were defined as CD123+ and monocytic AML cells as CD14+.

### Reporting summary

Further information on research design is available in the Nature Portfolio Reporting Summary linked to this article.

## Data availability

The publicly available data used in this study are available from Genomic Data Commons as study laml_2012 [https://gdc.cancer.gov/about-data/publications/laml_2012] (TCGA[9] SNV and indel data), from the cBioPortal for Cancer Genomics (https://www.cbioportal.org/datasets) as studies TCGA, NEJM 2013 [https://cbioportal-datahub.s3.amazonaws.com/laml_tcga_pub.tar.gz] and OHSU, Nature 2018 [https://cbioportal-datahub.s3.amazonaws.com/aml_ohsu_2018.tar.gz] TCGA[9] and Beat-AML1[31] gene expression data), from Vizome as study BeatAML [http://www.vizome.org/aml/geneset/] (Beat-AML1[31] SNV and indel data), from Github repositories BeatAML2 [https://biodev.github.io/BeatAML2/] (Beat-AML2[34] gene expression data) and AML-multistage [https://github.com/gerstung-lab/AML-multistage/tree/master/data] (Papaemmanuil et al.[10], SNV and indel data), and from Zenodo as study Clinseq_AML [https://zenodo.org/records/292986] (Clinseq[35] gene expression data). The raw sequencing data generated in this study have been deposited in the European Genome-Phenome Archive (EGA; https://ega-archive.org/) under the accession codes: EGAD50000001574 (https://ega-archive.org/datasets/EGAD50000001574; MP-WGS data), EGAD50000001575 (https://ega-archive.org/datasets/EGAD50000001575; WES data), EGAD50000001576 (https://ega-archive.org/datasets/EGAD50000001576; RNA-seq data), and EGAD50000001577 (https://ega-archive.org/datasets/EGAD50000001577; scRNA-seq data). The raw sequencing data deposited at EGA are available under restricted access due to the General Data Protection Regulation (GDPR), the Swedish data protection legislation, and the Swedish Public Access to Information and Secrecy Act. Access can be obtained by contacting request@researchdata.lu.se. Access will be granted for projects ensuring data protection in compliance with the aforementioned legislation, which can be further specified in a data access agreement provided upon request. The first response after requests for access is expected to occur within five business days. Once access has been granted, data will be available for the duration of the specified project. The processed gene expression data are available as gene expression matrices

from the SciLifeLab Data Repository (https://figshare.scilifelab.se/) through the following DOIs: https://doi.org/10.17044/scilifelab.21557163[72] (RNA-seq data), https://doi.org/10.17044/scilifelab.23715648[73] (scRNA-seq data). The remaining data are available within the Article, Supplementary Information, or Source Data file. Source data are provided with this paper.

## Code availability

Software and documentation for making KNN force plots and projecting single-cell data are available in the SingleCellProjections.jl package, at https://github.com/BioJulia/SingleCellProjections.jl. All custom code to reproduce the analyses supporting this paper is available at https://github.com/rasmushenningsson/AMLStateSpace, and also available through Zenodo (https://doi.org/10.5281/zenodo.17337903)[74] and as a Code Ocean capsule (https://doi.org/10.24433/CO.6447700.v1) [75].

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

## Acknowledgements

The authors thank the Center for Translational Genomics at Lund University and Clinical Genomics Lund, SciLifeLab, for providing single-cell sequencing service. The authors also thank the investigators of The Cancer Genome Atlas (TCGA) study on AML[9], the Beat AML program[31,34], the Clinseq study[35], and the study by Papaemmanuil et al[10], for sharing genomic and/or transcriptomic data used for validation of mutational profiles and differential gene expression analysis (data were accessed as detailed in supplemental Methods). This work was supported by the Swedish Childhood Cancer Fund (#PR2023-0097 to T.F.), the Swedish Cancer Society (#23-2953 PJ 01 to T.F.), the Swedish Research Council (#2023-02376 to T.F.), Governmental Funding of Clinical Research within the National Health Service (ALF-grant to T.F.), the Knut and Alice Wallenberg Foundation (#2022-0232 to T.F.), the Cancera Foundation (T.F.), the Mats Paulsson Foundation (T.F.), and Mrs. Berta Kamprad's Cancer Foundation, through a grant awarded to the L2CancerBridge program at CREATE Health Cancer Center (T.F.).

## Author contributions

H.L. and T.F. conceived the project. H.L., P.P.M., H.T., M.R., N.P.M., Sv.P., H.Å., C.S., and C.O.P. performed experiments. H.L., P.P.M., H.T., R.H., M.R., N.L., N.P.M., Sv.P., V.R., P.S., J.D., M.F., H.Å., C.S., C.O.P., and T.F. analyzed the data. N.L., P.S., X.Z., G.J., V.L., and S.L. provided samples and clinical data. H.L. and T.F. wrote the manuscript, which was reviewed and edited by the other co-authors.

## Funding

## Competing interests

The authors declare no competing interests.
