## [Transparent Peer Review file · Nature Communications]

The AML cellular state space unveils NPM1 immune evasion subtypes with distinct clinical outcomes

Corresponding Author: Dr Henrik Lilljebjörn

Version 0:

Reviewer comments:

Reviewer #2

(Remarks to the Author)

Lilljebjörn et al. have significantly revised the interpretation of the results. The initial manuscript attributed potential immune evasion to NPM1 class I due to downregulation of MHC class II, whereas the revised version associates NPM1 class II with resistance to allogeneic T cells. This attests to the complexity of the immune response in AML. The authors suggest that NPM1 class I evades anti-tumor immunity, whereas NPM1 class II engages in immune resistance or suppression, the latter leading to worse outcomes. This is an interesting model. The manuscript is a valuable contribution to the field.

The following additions should be available before publication:

- Supplementary table 2 should have a column with the efficiency of mutation calling for each mutation, i.e., the proportion of cells in which a wildtype or mutated transcript was confidently detected. This will help others in the community select efficient primers.
- I cannot evaluate the reproducibility or annotation of code since the authors have not made it available ("link will come")
- Data is not available (<https://doi.org/10.17044/scilifelab.23715648> DOI cannot be found). Since the authors used Seurat, it would be very helpful to share a Seurat object with count data and metadata for each cell, for example, using Figshare.

Reviewer #4

(Remarks to the Author)

In the revised paper, authors have adequately addressed the prior critiques raised by reviewer. The T cell responsive difference between NPM1-mutant class I and II patients has been more carefully studied with scRNA-seq data, flow and ex vivo cell co-culture system.

Reviewer #5

(Remarks to the Author)

The main finding of the project is the segregation of NPM1 mutated patients into two distinct subgroups based on single cell RNA-sequencing. Patients in the NPM1 Type II group have a significantly reduced outcome after allogeneic hematopoietic stem cell transplantation compared to patients in the NPM1 Type I group. This is an interesting and clinically relevant finding. The author claim that based on the in vitro data provided this may be due to resistance of the type II patients to allogeneic T-cell killing. However, the mechanism of resistance could not be fully elucidated. Of note, more patients in the type II group co-express mutations in DNMT3a and FLT3, which confer adverse outcome. Other factors suggested by the authors that may explain the difference are a heterogeneous immune checkpoint expression and a higher proportion of naive CD8 T-cells in the type II patients.

General comment:

The authors provide robust, extensive, and compelling additional data. However, the cohort is poorly characterized and the data needs to be confirmed in an independent cohort.

Specific comments:

1. The authors need to provide a table that compares the clinical and molecular features of NPM1 class I and class II patients with and without allo-HSCT. It should also include age and the ELN2022 risk category and not the ELN 2017 risk.
2. The outcome of patients with NPM1 mutations without allo-HSCT seems particularly poor, which relates to comment Nr.1. Since AML with NPM1 mutations without FLT3 mutations belong to the ELN2022 favorable risk group, the authors need to indicate the reason for allo-HSCT for these patients.
3. There is no information about treatment before allo-HSCT and treatment response to the initial therapy. Since the response before allo-HSCT is significantly associated with outcome it needs to be provided, ideally including MRD status (NPM1) before allo-HSCT.
4. The ELN risk is missing in 18(!) patients and needs to be provided. Please indicate ELN2022 risk for all patients.
5. There is 8 patients with APL (PML-RARA), who need to be excluded from the outcome analysis since they receive a completely different treatment (ATRA-based) and usually are not treated by allo-HSCT.
6. 12 patients with a NPM1 mutation belong to the ELN low risk group and typically are not treated by allo-HSCT. Can the authors indicate whether these patients belong to NPM1 class I or II?
7. Although the authors claim that their findings may be clinically relevant, and I agree with that, they should discuss how their findings could be translated into a clinical application. The 2 NPM1 subclasses differ in the expression of 180 genes. Can the author derived a clinically-applicable signature with only a few genes? Alternatively, could the 2 subclasses be distinguished by a flow cytometry panel based on the gene expression?
8. Lastly, the outcome data obtained needs to be validated in silico in an independent cohort (e.g., BEAT-AML) to strengthen the findings.

Version 1:

Reviewer comments:

Reviewer #5

(Remarks to the Author)

I would like to thank the authors for following my suggestions. The characterization of the cohort has significantly improved and my concern regarding inclusion APL has been clarified. Also, it is now clear that the data was validated independently.

The MRD data has been added, but may be difficult to understand in its present state. Can the authors indicate the method for MRD measurement (Flow, PCR, NGS)? Also, indicating the number of months after diagnosis is not ideal. For NPM1 mutated patients the MRD status after completion of induction therapy is correlated with outcome. Would it be possible to label the column e.g. MRD after induction therapy and indicate only one value? Since only few patients had MRD assessment before allo-HSCT I suggest to remove this column.

(Remarks on code availability)

Reviewer #2 (Remarks to the Author):

Lilljebjörn et al. have significantly revised the interpretation of the results. The initial manuscript attributed potential immune evasion to NPM1class I due to downregulation of MHC class II, whereas the revised version associates NPM1class II with resistance to allogeneic T cells. This attests to the complexity of the immune response in AML. The authors suggest that NPM1class I evades anti-tumor immunity, whereas NPM1class II engages in immune resistance or suppression, the latter leading to worse outcomes. This is an interesting model. The manuscript is a valuable contribution to the field.

Reviewer #2 (Comments and questions):

1. Supplementary table 2 should have a column with the efficiency of mutation calling for each mutation, i.e., the proportion of cells in which a wildtype or mutated transcript was confidently detected. This will help others in the community select efficient primers.

Response: Thank you for this suggestion. Two new columns have been added to supplementary table 2, detailing efficiency of mutation calling for each mutation. The first column details the proportion of cells in which a mutation was detected. The second column details the proportion of cells in which either wildtype or mutated transcripts were detected.

2. I cannot evaluate the reproducibility or annotation of code since the authors have not made it available ("link will come")

Response: All code is now made available, and a new code availability statement has been included in the manuscript:

Software and documentation for making knn force plots and projecting single cell data is available in the SingleCellProjections.jl package, at <https://github.com/BioJulia/SingleCellProjections.jl>. All custom code to reproduce the analyses supporting this paper is available at <https://github.com/rasmushenningsson/AMLStateSpace/>.

3. Data is not available (<https://doi.org/10.17044/scilifelab.23715648> DOI cannot be found). Since the authors used Seurat, it would be very helpful to share a Seurat object with count data and metadata for each cell, for example, using Figshare.

Response: All raw data produced within the project (WES, MP-WGS, RNA-seq and scRNA-seq) have now been deposited to the European genome-phenome archive (EGA; <https://www.ega-archive.org/>). In addition, processed gene expression data for the bulk RNA-seq and single cell RNA-seq datasets are available as count matrices and a Seurat object with both count data and metadata at the SciLifeLab Figshare Data Repository (<https://figshare.scilifelab.se/>).

This information has also been added to the data availability statement:

Source data are provided with this paper. In addition, raw sequencing data generated during the current study are available from the European Genome-Phenome Archive (EGA; <https://ega-archive.org/>) using the following accession numbers: EGAD50000001574 (MP-WGS data), EGAD50000001575 (WES data), EGAD50000001576 (RNA-seq data), and EGAD50000001577 (scRNA-seq data). The datasets are also available as gene expression matrices from the Scilifelab Data Repository (<https://figshare.scilifelab.se/>) through the following DOIs: <https://doi.org/10.17044/scilifelab.21557163> (RNA-seq data), <https://doi.org/10.17044/scilifelab.23715648> (scRNA-seq data).

Reviewer #4 (Remarks to the Author):

In the revised paper, authors have adequately addressed the prior critiques raised by reviewer. The T cell responsive difference between NPM1-mutant class I and II patients has been more carefully studied with scRNA-seq data, flow and ex vivo cell co-culture system.

Response: We would like to thank the reviewer for these positive remarks.

Reviewer #5 (Remarks to the Author):

The main finding of the project is the segregation of NPM1 mutated patients into two distinct subgroups based on single cell RNA-sequencing. Patients in the NPM1 Type II group have a significantly reduced outcome after allogeneic hematopoietic stem cell transplantation compared to patients in the NPM1 Type I group. This is an interesting and clinically relevant finding. The author claim that based on the in vitro data provided this may be due to resistance of the type II patients to allogeneic T-cell killing. However, the mechanism of resistance could not be fully elucidated. Of note, more patients in the type II group co-express mutations in DNMT3a and FLT3, which confer adverse outcome. Other factors suggested by the authors that may explain the difference are a heterogeneous immune checkpoint expression and a higher proportion of naive CD8 T-cells in the type II patients.

General comment:

The authors provide robust, extensive, and compelling additional data. However, the cohort is poorly characterized and the data needs to be confirmed in an independent cohort.

Reviewer #5 (Comments and questions):

1. The authors need to provide a table that compares the clinical and molecular features of NPM1 class I and class II patients with and without allo-HSCT. It should also include age and the ELN2022 risk category and not the ELN 2017 risk.

Response: The ELN2022 risk category has now been included in Supplementary Table 4 “Clinical and molecular features of AML cases with *NPM1* mutations”, which lists all *NPM1*-positive cases in the local dataset. Age, *NPM1* class and allo-HSCT status are now also provided. However, information regarding the ELN2017 risk

score has also been retained within the table for comparison. We have also added a new Supplementary Table 5 which lists this information also for the *NPM1*-positive cases in the external cohorts (to the extent that it was available to us).

2. The outcome of patients with *NPM1* mutations without allo-HSCT seems particularly poor, which relates to comment Nr.1. Since AML with *NPM1* mutations without *FLT3* mutations belong to the ELN2022 favorable risk group, the authors need to indicate the reason for allo-HSCT for these patients.

Response: We agree that it is important to consider the reason for allo-HSCT, for example for patients in the favorable risk group. Information regarding the reason for allo-HSCT has now been added to Supplementary Table 4 “Clinical and molecular features of AML cases with *NPM1* mutations”, which lists all *NPM1*-positive cases in the local dataset. We note that relapse is the most common reason for allo-HSCT in the favorable risk group.

3. There is no information about treatment before allo-HSCT and treatment response to the initial therapy. Since the response before allo-HSCT is significantly associated with outcome it needs to be provided, ideally including MRD status (*NPM1*) before allo-HSCT.

Response: Information regarding treatment before allo-HSCT is now also included, both in Supplementary Table 4 “Clinical and molecular features of AML cases with *NPM1* mutations” and Supplementary Table 5 “Clinical features of external AML cases with *NPM1* mutations”. These two tables cover all cases in the local dataset (Supplementary Table 4) and the external datasets (Supplementary Table 5). However, while all patients are included in these tables, only patients treated with curative intent (intensive treatment) were included in the survival analyses.

4. The ELN risk is missing in 18(!) patients and needs to be provided. Please indicate ELN2022 risk for all patients.

Response: Thank you for highlighting the lack of ELN risk data for a number of patients. All samples have now been classified according to ELN2022. This information is now presented in Figure 1A and Supplementary Table 1 “Clinical features of AML cases and molecular methods applied”. For *NPM1*-mutated cases, the ELN2022 risk group is also included in Supplementary Table 4 “Clinical and molecular features of AML cases with *NPM1* mutations” and Supplementary Table 5 “Clinical features of external AML cases with *NPM1* mutations”.

5. There is 8 patients with APL (PML-RARA), who need to be excluded from the outcome analysis since they receive a completely different treatment (ATRA-based) and usually are not treated by allo-HSCT.

Response: The outcome analyses presented in the manuscript only include patients with *NPM1*-mutated AML. Thus, the eight patients with APL that are included in the local RNA-seq dataset are already excluded from any outcome analysis. These cases were included to provide contrast in the genomic, gene expression, and cell distribution analyses.

6. 12 patients with a NPM1 mutation belong to the ELN low risk group and typically are not treated by allo-HSCT. Can the authors indicate whether these patients belong to NPM1 class I or II?

Response: The twelve cases classified in the ELN2022 risk group “Favorable” are evenly distributed between *NPM1* class I and *NPM1* class II, with six cases in each group. This information is now available in Supplementary Table 4 “Clinical and molecular features of AML cases with *NPM1* mutations”, that now includes information both on ELN2022 risk group and *NPM1* class.

7. Although the authors claim that their findings may be clinically relevant, and I agree with that, they should discuss how their findings could be translated into a clinical application. The 2 *NPM1* subclasses differ in the expression of 180 genes. Can the author derived a clinically-applicable signature with only a few genes? Alternatively, could the 2 subclasses be distinguished by a flow cytometry panel based on the gene expression?

Response: We thank the reviewer for raising this very relevant point. To explore the feasibility of identifying a clinically applicable gene signature smaller than the 180-gene signature used in the manuscript to identify *NPM1* class identity in RNA-seq data, we identified a reduced set of 30 genes, containing the ten most upregulated genes from each of the groups *NPM1* class I, *NPM1* class II, and unclassified cases. When all the cases were clustered based on this reduced set of genes, 315/370 cases (85%) were assigned the same *NPM1* subtype as they were when clustered based on the original 180-gene signature. Of the discordant cases, 49/55 were moved between being in the unclassified groups to *NPM1* class I or II, or vice versa. Since we speculate that the unclassified group encompasses cases with too few AML immature cells in the analyzed sample for classification into *NPM1* class I or II, the discrepancies could represent a difference in sensitivity between the two gene sets. Thus, we conclude that the most robust way to identify *NPM1* class I and II is using the original 180-gene signature with gene expression data from RNA-sequencing, which is increasingly used by diagnostic laboratories. However, our results also indicate that it should be possible to identify a smaller set of genes for use in a more restricted clinical assay, but that this would ideally include a dedicated effort based on data from a larger prospective series of cases analyzed by scRNA-seq, to be able to characterize the samples that currently fall into the unclassified group better (since no such samples were present in the local dataset). These new results are described in a new paragraph in the results section, page 11:

Given the potential clinical application of the novel NPM1 classification, we examined if a shorter gene list for distinguishing the classes could be identified, as this could enable a custom gene expression assay for clinical implementation. Hierarchical clustering based on a list of 30 genes, encompassing the ten most overexpressed genes in each of $NPM1^{class I}$, $NPM1^{class II}$ and the unclassified cases, produced three distinct clusters in all datasets (Supplementary Fig. 15; Supplementary Table 6). In total, this method identified the correct NPM1 class in 315/370 cases (85%). Most misclassifications occurred to or from the unclassified group (49/55 discordant cases), suspected to encompass cases with too few AML immature cells for classification based on bulk gene expression data. Thus, this indicates that it is possible to develop a custom assay for clinical identification of NPM1 class, but that

this should ideally involve a dedicated effort including scRNA-seq data from a larger cohort to also be able to characterize the unclassified group.

8. Lastly, the outcome data obtained needs to be validated in silico in an independent cohort (e.g., BEAT-AML) to strengthen the findings.

Response: We agree that independent in-silico validation is important. Indeed, the results were validated in independent cohorts, as detailed below. We apologize that this was not sufficiently well described in our original manuscript. To clarify the methodology and to highlight the robustness of this finding, we have now modified Supplementary Figure 14 to show more clearly that these results were first identified in a discovery cohort containing the local Lund data (n=28 cases), the TCGA dataset (n=40 cases) and the dataset from the first Beat-AML publication (ref 31; n=88 cases). The results were then validated in a cohort consisting of the Clinseq dataset (n=102 cases) and the new data published in the second Beat-AML publication (ref 34; n=47 cases; all distinct from those in the discovery cohort). Importantly, the inferior survival following allo-HSCT for patients with *NPM1* class II AML is also apparent in all datasets individually, as shown in Supplementary Figure 14.

Reviewer #5 (Remarks to the Author):

I would like to thank the authors for following my suggestions. The characterization of the cohort has significantly improved and my concern regarding inclusion APL has been clarified. Also, it is now clear that the data was validated independently.

The MRD data has been added, but may be difficult to understand in its present state. Can the authors indicate the method for MRD measurement (Flow, PCR, NGS)? Also, indicating the number of months after diagnosis is not ideal. For NPM1 mutated patients the MRD status after completion of induction therapy is correlated with outcome. Would it be possible to label the column e.g. MRD after induction therapy and indicate only one value? Since only few patients had MRD assessment before allo-HSCT I suggest to remove this column.

Response: Thank you for the suggestions on how to improve the presentation of the MRD data. The Supplementary Data 4 table has now been modified, incorporating all changes suggested by the reviewer: MRD data is now presented in a single column labeled "MRD after induction (%)", containing a single relevant value for each patient. In addition, the MRD methodology is further detailed in a footnote explaining that MRD was performed using flow cytometry for all cases.